



**A sea surface temperature reconstruction for the southern Indian**
**Ocean trade wind belt from corals in Rodrigues Island (19°S, 63°E)**
J. Zinke[1,2,3], L. Reuning[4], M. Pfeiffer[4], J. Wassenburg[5], E. Hardman[6], R. Jhangeer-Khan[6],
Davies, G. R.[7], C.K.C. Ng[8], and D. Kroon[9]
[1]Department of Environment and Agriculture, Curtin University of Technology, Kent
Street, Bentley, WA6102, Australia
[2]Australian Institute of Marine Science, Nedlands, WA 6009, Australia
[3]School of Geography, Archaeology and Environmental Studies, University of
Witwatersrand, Johannesburg, South Africa.
[4]Institute for Geology, RWTH Aachen, Wuellnerstrasse2, 52056 Aachen, Germany
[5]Institute for Geosciences, Johannes-Gutenberg-University Mainz, Johann-Joachim-
Becher-Weg 21, D-55128 Mainz
[6]SHOALS Rodrigues, Rodrigues, Mauritius
[7]Department of Petrology, VU University Amsterdam, De Boelelaan 1085, 1081 HV
Amsterdam, Netherlands
[8]Department of Medical Radiation Sciences, Curtin University of Technology, Kent
Street, Bentley, WA6102, Australia
[9]University of Edinburgh, School of GeoSciences, The King's Buildings, West Mains
Road, Edinburgh EH9 3JW, UK.
**Correspondence to:** Jens Zinke, jens.zinke@gmail.com



**Abstract**
The western Indian Ocean has been warming rapidly over the past decades and this has
adversely impacted the Asian Monsoon circulation. It is therefore of paramount
importance to improve our understanding of links between Indian Ocean Sea Surface
Temperature (SST) variability, climate change, and sustainability of reef ecosystems.
Here we present two monthly-resolved coral Sr/Ca records (Totor, Cabri) from Rodrigues
Island (63°E, 19°S) in the south-central Indian Ocean trade wind belt, and reconstruct
SST based on the linear relationship with the Sr/Ca proxy. The records extend to 1781
and 1945, respectively. We assess the reproducibility of the Sr/Ca records, and potential
biases in our reconstruction associated with the orientation of corallites. We quantify
long-term SST trends and identify interannual relationships with the El Niño-Southern
Oscillation (ENSO) and the Pacific Decadal Oscillation (PDO). We conclude that careful
screening for diagenesis and orientation of corallites is of paramount importance to assess
the quality of Sr/Ca-based SST reconstructions. Our proxy records provide a reliable SST
reconstruction between 1945 and 2006. We identify strong teleconnections with the
ENSO/PDO over the past 60 years, eg. warming of SST during El Niño or positive PDO.
We suggest that additional records from Rodrigues Island can provide excellent records
of SST variations in the southern Indian Ocean trade wind belt and teleconnections with
the ENSO/PDO on longer time scales.
**1 Introduction**
The Indian Ocean has been warming steadily over the past century with the western
portion of the basin having experienced an increase in SST of up to 1.2°C over the past



60 years (Koll Roxy et al., 2014). The Indian Ocean has also taken up a large amount of
heat in its interior during the recent 15 years when global SST increased at a smaller rate
as compared to previous decades (Lee et al., 2015). The strong Indian Ocean warming
over the past century is thought to have contributed to a decreasing land-sea thermal
contrast with the Indian subcontinent affecting monsoon rainfall and might have played a
major role for the decrease in East African rainfall between March to May in recent
decades (Funk et al., 2008; Koll Roxy et al., 2015). The western Indian Ocean warming
has also been shown to follow closely anthropogenic radiative forcing over the past
century (Funk et al., 2008; Alory et al., 2009; Koll Roxy et al., 2015). Furthermore, the
western Indian Ocean warmed significantly during past El Niño events with the 1997/98
event having caused widespread coral bleaching and mortality. It is therefore of
paramount importance to improve our understanding of links between Indian Ocean SST
variability,  global climate change, and sustainability of reef ecosystems. Yet, long-term
observational records of Indian Ocean SST are sparse and are thought to be only reliable
after the 1960's (Tokinaga et al., 2012). To overcome the limitations of the short
observational record, paleoclimate records of past SSTs can be generated to provide
insight into long-term SST changes and interannual to decadal variability.
Paleoclimate reconstructions of SST from massive corals have provided
invaluable records for past SST trends and interannual to decadal variability in the
western Indian Ocean (Charles et al., 1997; Cole et al., 2000; Cobb et al., 2001; Pfeiffer
et al., 2004, 2009; Pfeiffer & Dullo, 2006; Nakamura et al., 2009; Crueger et al., 2009;
Grove et al., 2013; Zinke et al. 2008, 2009, 2014). Massive corals, such as *Porites* spp.,
can grow for centuries and grow at a rate between 0.5 and 2 cm yr$^{-1}$. Therefore, down-



core sampling of massive corals yields an in situ SST time series of monthly resolution.
As the coral precipitates its skeleton, trace elements and stable isotopes are incorporated
at different concentrations, relative to Ca, in relation to changing SSTs (Felis and Pätzold,
2003). Both, the Sr/Ca ratio and $\delta^{18}$O composition of the coral aragonite were shown to
be reliable paleo-thermometers, whereby a negative relationship exists with SST (Alibert
and McCulloch, 1997; Pfeiffer & Dullo, 2006; DeLong et al., 2012). A compilation of
Sr/Ca-SST calibrations for Porites *spp.* revealed a mean Sr/Ca relationship with SST of -
0.061mmol/mol/1°C SST increase (Corrège, 2006). Since Sr has a long oceanic residence
time, skeletal Sr/Ca is assumed to mainly reflect SST variability. The quality and
accuracy of paleo-thermometers strongly depends on optimal sampling of the major
growth axes (De Long et al., 2012). Furthermore, diagenetic alterations of coral aragonite
can lead to errors in SST reconstructions and need to be excluded by specific analysis
(McGregor and Gagan, 2003; McGregor and Abram, 2008; Sayani et al., 2011; Smodej et
al., 2015).

Currently, none of the coral proxy records from the western Indian Ocean cover

the south-central Indian Ocean basin in the heart of the trade wind system. Furthermore,
all proxy records of interest for the trade wind belt are based on oxygen isotopes with the
exception of two Sr/Ca ratio records covering 1952 to 2008 from St. Marie Island off
East Madagascar (Grove et al., 2013). The latter provided mixed results with
discrepancies in terms of the long-term SST trend estimates due to confounding effects of
coral calcification in at least one of the cores (Grove et al., 2013). A coral oxygen isotope
record from Reunion Island (21°S, 55°E; Mascarene Islands) located approximately
230km to the southwest of Mauritius spans the period 1832 to 1994 and is the longest for



the subtropical region off East Madagascar (Pfeiffer et al., 2004). Pfeiffer et al. (2004)
showed evidence that the La Reunion coral dominantly recorded past variations in the
salinity anomalies associated with transport changes of the South Equatorial Current. The
proxy record showed decadal anomalies that were opposite to those of SST. Crueger et al.
(2009) showed close linkages of the salinity, SLP and SST signal associated with the
Pacific Decadal Oscillation (Mantua et al., 1997) in coral records from Reunion and Ifaty
(SW Madagascar), respectively. Two coral oxygen isotope records from the Seychelles
located in the tropical western Indian Ocean (5°S, 54°E) were interpreted as an excellent
record of past Southwest Monsoon SST changes and showed strong correlations with air
temperatures over India between 1847 to 1994 (Charles et al., 1997; Pfeiffer & Dullo,
2006). Both, the Reunion and Seychelles records showed strong correlations with the El
Nino-Southern Oscillation on interannual and decadal time scales (Pfeiffer & Dullo,

2006).

Here, we aim to reconstruct past SSTs from Sr/Ca ratios in two coral cores

obtained from Rodrigues Island (19°S, 63°E) located 500 km to the North-East of
Mauritius within the trade wind belt of the south-central Indian Ocean. We assess the
reproducibility of the Sr/Ca proxy from two different locations, their long-term trends and
interannual variability related to the El Nino-Southern Oscillation.

**2 Regional setting and climate**
Rodrigues (63°E, 19°S) is a small volcanic island in the southern Indian Ocean, about
619 km east of Mauritius (Fig. 1). It is part of the eastern edge of the Mascarene Plateau
that is made up of Lower Tertiary basalts (Mart 1988) that formed by a seaward flow of



lava, which has been eroded by hydrodynamic forces, and biological and chemical
processes (Turner and Klaus, 2005). Rodrigues has a surface area of about 119 km$^2$, with
a maximum altitude of 396 meter above sea level and is surrounded by a nearly
continuous fringing reef, which form an almost continuous band measuring
approximately 90km in length (Turner and Klaus, 2005; Lynch et al. 2002). The reef
encloses a shallow lagoon, which, at 240km$^2$, is twice the area of the island itself. The
maximum tidal range is approximately 1.5m, and since the average water depth in the
lagoon is less than 2m, some areas are exposed at low spring tides. The water depth
immediately beyond the reef slopes is usually within the range of 10m to 30m. The island
has three major channels, one dredged channel for the main harbour at Port Mathurin in
the north, and natural channels in the south near Port Sud Est and in the East at St
Francois. Several small passes are also found at intervals around the reef (Turner and
Klaus, 2005).
The water surrounding Rodrigues is supplied by the South Equatorial Current (SEC)
(New et al. 2007) which is a broad east to west current between 10º and 20º S in the
Indian Ocean driven by the southeast trade winds (Schott and McCreary, 2001). The
southern part of the SEC water flows in several directions, alongside Rodrigues in
southwest and southeast direction, and westward to Mauritius (New et al. 2007).
Rodrigues has a relatively dry climate and evaporation exceeds precipitation on the
annual mean. Yearly precipitation is ~1000 mm with most precipitation from January to
April related to the position of the Inter Tropical Convergent Zone (ITCZ). Between
November and March, the Southern Indian Ocean is affected by tropical cyclones, as a
result of warm SSTs and a strong convergence between northeast and southeast trades.



Rodrigues experiences two to sixteen cyclones per year, of which 2.5 are extreme: with
winds of 280 km/h and waves that reach 100 m inland and 2 m above sea level. They
usually last five to ten days (Turner and Klaus, 2005).
SST was monitored *in situ* by a CTD 150m offshore from the northern fringing reefs
at Totor between 2002 to 2006 (Fig. 2a). Maximum SST are recorded between December
to March (28.6 ± 0.5ºC) and minimum SST between July to September (22.4 ± 0.27ºC).
Annual mean SST is 25.49 ± 0.24ºC with a seasonal amplitude of 6.22 ± 0.68ºC (Fig. 2a).
Air temperatures are recorded by the WMO weather station located at the northern
coast of Rodrigues since 1951 and are available at http://climexp.knmi.nl/. The most
recent years between 1997 and 2007 have been provided by the Rodrigues
Meteorological Office (Fig. 2b). The warmest months are December to March (31.2 ±
0.3º), the coldest months are July to September (24.2 ± 0.3º). Yearly average air
temperature is 27.49ºC ± 0.31ºC with a yearly amplitude of about 7 ± 0.79ºC.

**3 Materials and Methods**
Two cores were drilled from massive, dome-shaped *Porites* sp. and *Porites lobata*
at the northern reef sites Totor (S19º67062; E63º42923) and Cabri (S19°667171,
E63°43.4423), respectively (Fig. 1; Table 1). The size of the coral colonies at Totor is
~2.5m and that of Cabri is ~4m in height. Both colonies were healthy and showed no
signs of disease or dead surfaces at the time of drilling. The 220cm long core Totor was
obtained in August 2005 from the forereef slope of the northern fringing reef facing the
open ocean with the top of the colony at 4m water depth. The 180cm long core Cabri was
obtained in March 2007 growing in 3m water depth about 1km to the northeast of Totor





from the outer fringing reef. The site Cabri is more exposed to trade winds as compared
to Totor which is more sheltered.

A commercially available pneumatic drill driven by scuba tanks was used to

extract cores along the central growth axis, with a diameter measuring 4 cm. Cores were
sectioned into 7 mm thick slabs, rinsed several times with demineralised water, blown
with compressed air to remove any surficial particles and dried for more than 24 hours in
a laminar flow hood. Growth laminae were visualised by X-radiograph-positive prints,
and the growth axis of the coral slab was defined as the line normal to these laminae
(Appendix Fig. 7 and 8). Coral densities ($g/cm^3$) were calculated by analysing digital X-
rays using the program CoralXDS and densitometry (Helmle et al., 2011; Carricart-
Ganivet et al., 2007), calcification rate ($g/cm^2\ yr^{-1}$) by multiplying density with extension
rate. The annual extension rates ($cm\ yr^{-1}$) were calculated by measuring the distance (cm)
between density minima using the program CoralXDS. With a diamond coated drill
mounted on top of a movable frame, samples were taken every 1 mm parallel to the
growth axis, equivalent to approximately monthly resolution.

A combination of X-ray images, X-ray diffraction (XRD), light and scanning

electron microscopy (SEM) with Energy Dispersive X-Ray Spectrometer (EDS) was used
to investigate possible diagenetic alteration from cores Totor and Cabri. All coral slabs
from cores Toto and Cabri were initially screened for diagenetic alterations using X-ray
images (Figs. A7, A8). Corals that showed an annual density banding without anomalous
high or low density patches were selected for further study and considered free from
obvious diagenetic alteration. Representative samples were chosen from both cores based
on the X-ray images for SEM, thin-section and XRD analysis. Additional samples were





selected after geochemical analysis targeting intervals with unusually high or low Sr/Ca
ratios. The powder-XRD diffractometer at RWTH Aachen University was calibrated to
detect and quantify very low calcite contents (detection limit ~ 0.2%) following the
method described by Smodej et al. (2015). In addition, the 2D-XRD system Bruker D8
ADVANCE GADDS was used for XRD point-measurements directly on the coral slab
with a spatial resolution of ~ 4 mm and a calcite detection limit of ~ 0.2% (Smodej et al.,
2015). A 2-dimensional detector allows the simultaneous data collection over a large 2 θ
range, which reduces the counting time to 10 min for each sampling spot. The coral is
mounted on a motorized XYZ-stage and the position of each sample spot is controlled by
an automated laser-video alignment system. Multiple sample points can be predefined
and measured automatically. This method was used to test for the presence of secondary
calcite along the geochemical sample traces of both corals.

Sr/Ca ratios were measured at the University of Kiel with a simultaneous

inductively coupled plasma optical emission spectrometer (ICP-OES, Spectro Ciros CCD
SOP; Zinke et al., 2014). Approximately 0.5mg of coral powder are dissolved in 1.00 ml
0.2M $HNO_3$. Prior to analysis, this digest solution is diluted with 0.2M $HNO_3$ to a final
concentration of approx. 8ppm Ca. An in-house coral powder standard (Mayotte) was
prepared in an analogue way and used as consistency standard, being re-analyzed after
every six samples. The international reference material JCp-1 (coral powder) was
analyzed with every sample batch. All calibration solutions are matrix-matched to 8 ppm
Ca. Strontium and Ca are measured at their 407 and 317 nm emission lines. Our intensity
ratio calibration strategy combines the techniques described by de Villiers et al. (2002)



and Schrag (1999). Analytical precision of Sr/Ca determinations as estimated from
replicate measurements of unknown samples is 0.15% or 0.01 mmol/mol (1sigma).

The coral core chronologies were developed based on the seasonal cycle of Sr/Ca.

We assigned the coldest month (either August or September) to the highest measured
Sr/Ca ratio in any given year, according to both *in situ* SST and grid-SST (Extended
reconstructed SST; Smith et al., 2008). We then interpolated linearly between these
anchor points to obtain age assignments for all other Sr/Ca measurements. In a second
step, the Sr/Ca data were interpolated to 12 equidistant points per year to obtain monthly
time series using AnalySeries 2.0 (Paillard et al., 1996). This approach creates a non-
cumulative time scale error of 1 - 2 month in any given year, due to interannual
differences in the exact timing of peak SST. The monthly interpolated Sr/Ca time series
were cross-checked with the chronologies from coral XDS to reveal the timing of high
and low density banding. High density bands in both corals formed in summer (low
Sr/Ca) of any given year.

**4 Historical SST data**

Historical SST data collected primarily by ships-of-opportunity have been summarised

in the comprehensive ocean atmosphere data set (ICOADS) to produce monthly averages
on a 2°x2° grid basis (Woodruff et al., 2005). In the grid that includes Rodrigues Island
the data are extremely sparse (http://climexp.knmi.nl). We therefore extracted SST from
extended reconstructed SST (ERSST version 3b/v4; Smith et al., 2008), also based on
ICOADS data, which uses sophisticated statistical methods to reconstruct SST in time of
sparse data. From ERSST, we extracted data in the 2°x2° grid centred at 61-63°E, 19-



21°S (Table A1). Between 2002 and 2006 (*in situ* data coverage) ERSST version 3b
shows a yearly average of about 25.57C ± 0.19ºC with a yearly amplitude of 5.14 ±
0.39ºC (Smith et al., 2008). The warmest months are February and March with a SST of
28.29ºC ± 0.4ºC, the coldest months are August and September with a SST of 23.15ºC ±
0.13ºC.

Furthermore, we used Met Office Hadley Centre's sea ice and sea surface temperature

(HadISST) data for the grid 62-63°E, 19-20°S (Rayner et al., 2003; Kennedy et al., 2011;
Table A1). HadISST temperatures are reconstructed using a two-stage reduced-space
optimal interpolation procedure, followed by superposition of quality-improved gridded
observations onto the reconstructions to restore local detail. Since January 1982, SST
time series for HadISST use the optimal interpolation SST (OISST; 1°x1°), version 2
(Reynolds et al., 2002) that includes continuous time series of satellite-based SST
measurements. We also extracted Advanced Very High Resolution Radiometer
(AVHRR) SST at 0.25°x0.25° resolution (Reynolds et al., 2007) from 1985 to 2006
which is also used by NOAA's coral reef watch. AVHRR SST for Rodrigues between
2002 and 2006 (*in situ* data coverage) provided from NOAA at 0.25°x0.25° resolution
(Reynolds et al., 2007) shows a yearly average of 25.4 ± 0.11ºC with a yearly amplitude
of 5.9 ± 0.58ºC. Warmest SSTs are observed between January and March (28.65 ±
0.44ºC) and coolest SST between July to September (22.75 ± 0.21ºC).

SST from the 5°x5° HadSST3, the most sophisticated bias-corrected SST data to

date, were downloaded for the region 60-65°E, 15-20°S (Kennedy et al., 2011; Appendix
Table 1). Yet, HadSST3 contains data gaps throughout the record due to strict quality
control. SST is reported as anomalies relative to the 1961 to 1990 mean climatology.





In addition, we extracted 5°x5° nigh-time marine air temperature data from
HadMAT1 and HadNMAT2 datasets (Kent et al., 2013). HadNMAT2 contains data gaps
throughout the record due to strict quality control. Night-time marine surface air
temperature is highly correlated with SST but free of the biases introduced by changes in
SST measurement techniques (Tokinaga et al., 2012).

**5 Results**
**5.1 Coral growth parameters**
The average growth rate of the corals Totor (224 years) and Cabri (130 years)
over all years of growth were 9.82±0.19mm $y^{-1}$ and 11.79±0.25mm $y^{-1}$, respectively
(Table 1; Fig. A1). The Cabri core shows a growth disturbance at 1907 that led to partial
colony death. This is confirmed by three additional cores taken from this colony at
different angles which all showed the mortality event marked by a dead surface pre-
dating ~1907. This lower core section is overprinted by diagenesis and it is therefore not
suitable for climate studies or to determine density and calcification rates.
Extension rate of the Cabri coral shows no long-term trend, yet shows high
interannual and decadal variability (Fig. A1). The same holds for calcification rates. Both
extension and calcification show marked interannual oscillation in the recent 10 years.
Skeletal density shows multidecadal oscillations with high densities between 1907 and
1935, the early 1940's, between 1958 and 1966 and 1980 and 2006, with lower densities
in between (Fig. A1).
The Totor core shows a similar decadal and interannual variability in extension
and calcification compared to the Cabri core for the period of overlap between 1877 and





2005 (Fig. A1). The fit is less optimal between 1877 and 1907 due to the dead surface in
Cabri that has obscured density banding. No significant trend is observed in both
extension and calcification rates over the entire record length. Skeletal density differs
between the two cores. The Totor core shows multi-decadal cycles in density
superimposed on a decreasing trend and larger magnitude density anomalies compared to
the Cabri core. Between 1960 and 2005 both density profiles agree well in terms of
decadal variability, both showed a significant drop since the late 1960's and recovery
thereafter. However, the low density period in the Totor core lasted several years longer.

**5.2 Seasonality, trends and variability in Sr/Ca and instrumental SST time series**

For the period of overlap between both cores (1945 to 2005) there is a between

colony offset in mean Sr/Ca of 0.0242 mmol/mol. Both cores show a distinct seasonality
in Sr/Ca throughout their record length (Fig. 3). The seasonality in the Totor core
(0.283±0.049 mmol/mol) is on average slightly higher compared to the Cabri core
(0.238±0.055 mmol/mol), yet both overlap within 1σ.

To eliminate the offset between Sr/Ca time series we calculated Sr/Ca anomalies

by subtracting their mean relative to the 1961 to 1990 reference period (Figure 3). We
subsequently calculated relative changes in SST based on the established empirical
relationship of -0.0607 mmol/mol per 1°C derived from >30 published Sr/Ca calibrations
(Corrège, 2006). A composite coral temperature record was then constructed by (1)
converting each proxy record to temperature units, (2) calculating the arithmetic mean of
the coral records from each site, and (3) averaging the mean records from both sites.



Between 1945 and 2006 both cores indicate higher Sr/Ca anomalies (a period of
cooling) that started in the mid 1950's and lasted until the early 1970's. Both cores show
a pronounced trend to more negative Sr/Ca values (warming) starting in the 1970's and
reduced seasonality in that period (Fig. 3). After 1984 Sr/Ca in the Cabri core further
decreases (warms) while core Sr/Ca in the Totor core has no trend. The detrended Sr/Ca
time series indicated that both cores show similar decadal oscillations between 1945 and
2005 (not shown). This highlights that the long-term trend estimates after 1984 need to be
viewed with caution.
The Sr/Ca time series in the Totor core extends to 1781. Marked negative Sr/Ca
anomalies (warmer) are observed during the first half of the 20$^{th}$ century centered at
1918/19, 1936-41 and in the period 1947-1951 that exceed anomalies in the 1961 to 1990
reference period. Sr/Ca anomalies between 1850 and 1900 are higher (cooler) while
decadal periods with lower (warmer) Sr/Ca are observed between 1781 and 1850 relative
to 1961 to 1990. The long-term trend in Sr/Ca anomalies between 1781 and 2005
converted to SST indicated an overall warming of 0.44°C.
The composite Sr/Ca time series displays interannual and decadal variability
throughout the record between 1781 and 2006. The anomaly around 1918/19 is the lowest
(warmest) of the entire record length. In general, Sr/Ca anomalies during the 20$^{th}$ century
are lower (warmer) than between 1850 and 1900, while anomalies between 1781 and
1850 reach similar levels relative to the period 1961 to 1990 for several decades with
short-lived excursions to higher (cooler) anomalies. The long-term trend in Sr/Ca
anomalies between 1781 and 2006 converted to SST indicated an overall warming of
0.37°C.




### 5.3 Calibration/validation of coral Sr/Ca-SST


We calibrated the coral Sr/Ca from both cores with *in situ* SST, ERSSTv.3b and
AVHRR SST for the period 2002 to 2006 using the minima and maxima in any given
year, as well as monthly values with AVHRR SST for 1981 to 2006 (Fig. 4; Tab. A2).
The slopes of the ordinary least squares regressions vary between -0.0384 to -0.0638
mmol/mol per 1°C (Tab. A2). The lowest slopes are obtained with *in situ* SST and the
highest with ERSSTv.3b (Tab. A2). We reconstructed absolute SST for the period of
overlap with *in situ* SST from 2002 to 2006 from both coral cores (Fig. 4). The Sr/Ca-
SST in the Totor core shows the best fit with *in situ* SST in terms of the seasonal
amplitude. The Sr/Ca-SST in the Cabri core overestimates the winter SST of 2002 and
2005, yet agrees well for 2003 and 2004 (Fig. 4). However, taking into account the
uncertainties (measurement error, regression error) around absolute SST from Sr/Ca for
Cabri and Totor of 1.23°C and 1.05°C (1σ), respectively, the coral data agree with *in situ*
SST within the 1σ uncertainty.
To eliminate large errors associated with absolute SST reconstructions from coral
Sr/Ca we calculated relative changes in SST for the composite coral temperature record
relative to the 1961 to 1990 mean based on the established empirical relationship of -
0.0607 mmol/mol per 1°C derived from >30 published Sr/Ca calibrations (Corrège,
2006). This slope is well within the range of our regressions based on a variety of SST
datasets (Tab. A2). We use a conservative estimate for the uncertainty around relative
SST changes based on the difference between lower (-0.04) and upper slope (-0.084)





estimates from these regression equations, thus ± 0.02 mmol per 1°C or ±0.33°C
(following Gagan et al., 2012; Tab. A2).

We validated the coral derived annual mean SST reconstruction against local Air

Temperature (AT), ERSSTv3b, ERSST4, HadISST, HadSST3 HadMAT1 and
HadNMAT2 for the period 1951 to 2006 (Figure 5; Figs. A4 to A6; See Supplementary
Tables 1-24 for mean annual correlations). The composite coral SST record clearly
follows instrumental SST in the grid box surrounding Rodrigues Island while the best fit
is obtained with local Rodrigues AT. Discrepancies with gridded SST products are
observed between 1951 and 1955. However, AT agrees with coral composite SST in that
period, yet not with core Cabri which tracks grid-SST between 1951 and 1955. However,
taking into account the uncertainty of ±0.33°C based on the regression error, coral
composite SST agrees with gridded SST within 1σ.

For the period 1951 to 2005, we used AT, ERSSTv3b, ERSST4, HadISST,

HadSST3, HadMAT1 and HadNMAT2 to validate trends in annual mean coral Sr/Ca-
SST anomalies (Fig. 5, 6). The long-term trends in Sr/Ca-derived SST anomalies for the
period 1951 to 2005 for Cabri and Totor converted to SST, using the published Sr/Ca-
SST relationship of -0.0607mmol/mol per 1°C, indicate a warming of 1.38±0.39°C and
cooling of -0.49±0.41°C, respectively. The composite Sr/Ca anomaly time series for 1951
to 2005 display a warming trend of 0.44±0.37°C. The uncertainty for the trend estimates
in coral Sr/Ca SST is calculated from the square root of the sum of squares of the
regression error and the error in the slope of the Sr/Ca-SST relationship. Instrumental
SST indicate a warming trend of 0.61±0.13°C for HadISST, 0.72±0.11°C for ERSST3b
(0.86±0.12°C for ERSST4) and 0.78±0.12°C for HadSST3. Air Temperature at



Rodrigues weather station recorded a warming trend of 0.46±0.17°C. All trends are
significant at the 2% level with the exception of the negative trend in Sr/Ca SST
anomalies in the Totor core which is not significant.
For the pre-1945 period we used ERSSTv3b, HadSST1 and HadSST3 to validate
annual mean coral Sr/Ca-SST back to 1854 and 1870, respectively (Figure 6). We stress
that the number of SST observations in the ICOADS SST database is extremely sparse
for our region (Fig. A2). However, the composite coral SST record tracks SST variations
for most of the past 150 years (Figure 6). The composite coral SST time series,
essentially the time series of core Totor, displays higher SST anomalies compared to all
gridded SST reconstructions in the 1850's, between 1916-1921, 1936-1941 and 1948-
1951 and lower SST anomalies for brief periods between 1850 and 1890. In general, the
coral composite SST is a valid reconstruction for the region surrounding Rodrigues Island
with the possible exception of 1854-1860, 1916-1921, 1936-1941 and 1948-1951 (Figure

6).

Largest discrepancies between grid-SST (starting from year 1854) and coral SST
reconstructions are found for core Totor with warm anomalies in the periods 1854-1860,
1916-1921, 1936-1941 and 1948-1951 (Figure 6). Interestingly, the correlation between
Totor Sr/Ca-SST, which dominates the coral composite time series pre-1945, has
significant correlations with HadSST3 (r=0.24; p=0.05; N=65) and HadNMAT2 (r=0.3;
p=0.014; N=64) observational time series only. The cool bias in coral derived SST
between 1882 and 1887 (slab 7) is most probably related to diagenetic alterations, but
none of the anomalously warm periods can be explained by diagenesis (see next section).
We assessed the orientation of corallites to the coral slab surface to test for sampling





artifacts that might have altered our Sr/Ca data which we summarized in Tables 2 and 3,
illustrate in Figure 7 and discuss in section 6.1. Most anomalous warm periods show sub-
optimal orientation of sampling path with corallites at an angle to the slab surface (see

6.1).


**5.4 Diagenetic tests for alterations of Sr/Ca profiles**
Representative samples for diagenetic screening with XRD, SEM and light
microscopy were identified on the coral slabs using the X-radiographs. Additionally,
intervals with presumably anomalous proxy values (warm or cold anomalies) were
analyzed with the same methods. Ten thin-sections, six SEM samples, ten powder-XRD
and thirteen spot-2D-XRD samples were analyzed from coral core Totor (Fig. 8). For
coral core Capri, seven thin-sections, one powder-XRD and six 2D-XRD samples were
analyzed. Neither powder nor spot-XRD analysis detected any calcite. Thin-section
analysis indicates a growth break within slab 12 that is also apparent in the radiograph
(Fig. 8; Fig. A7). Close to this break the coral is strongly affected by bioerosion and
encrustation by red algae (Fig. 8ef). However, the sampling transect for geochemical
analysis excluded this area and is therefore not affected by diagenesis (Fig. 8f).
Combined SEM, EDS and XRD analysis shows low amounts of patchy distributed
isopachous (~2μm) fibrous aragonite cement in slabs Totor 6 (1916-1921), 7 (1882-1887)
and 11 (~ 1809).
Aragonite cement should lead to higher Sr/Ca values and lower reconstructed
temperatures (Hendy et al., 2007). An interesting outcome is that the observed diagenesis
is not able to explain the Sr/Ca ratios except for the slab Totor 7. Here the observed



aragonite cement fits to relatively high Sr/Ca values resulting in a cold anomaly. No
anomalously high Sr/Ca ratios are associated with the patchy aragonite cements in slabs 6
and 11. Instead slabs 6 and 11 are characterized by low Sr/Ca ratios resulting in relatively
warm reconstructed temperatures. All other samples from the slabs Totor 3, 4, 8, 9 and 10
are devoid of diagenetic alteration. In summary, an influence of diagenesis on the proxy
record and resulting SST reconstructions can only be assumed for sample Totor 7 (years
1882-1887). Core Cabri showed only localized (single month) positive Sr/Ca anomalies
(cool SST bias). Thin-section and XRD analysis did not indicate any diagenetic
alteration, but the coral locally contained aragonitic sediment partially filling pore spaces
(Tab. 3). This aragonitic sediment potentially could have caused the isolated Sr/Ca peaks.
These individual data points were omitted from further analysis.

**5.5 Large scale teleconnections on interannual time scales**

For the period of most reliable data coverage between 1951 and 2006, the

detrended coral composite and Cabri Sr/Ca-SST records shows positive correlations for
austral summer and annual means with Indian Ocean wide SST and a positive correlation
with the central and eastern Pacific SST typical for the spatial ENSO and PDO pattern
(Figure 9; Supplementary Tables 24-25). We used HadISST (1870 and 2006) and
HadMAT1 to evaluate the long-tern spatial correlation pattern (Figs. A3 to A5). A similar
pattern emerged as for the period 1951 to 2006, yet of weaker magnitude across the
Pacific and confined to the southwestern Indian Ocean. We broke down the correlations
into 30 year segments starting in 1870 to test if the correlation changes throughout the
past 136 years. The ENSO/PDO pattern for austral summer is strong in the periods 1870-





1900, 1961-1990 and 1971-2006 (Fig. A3). Between 1900 and 1930 the correlation is not
significant. The large-scale teleconnections with SST are stronger for the Cabri Sr/Ca-
SST time series after 1945 (Figs. A4, A5), while core Totor has weaker and statistically
non-significant correlations in that period. This indicates that the Cabri time series is
more reliable for the recent 60 years for monthly averages and annual means and shows
the strongest correlations across the Indo-Pacific (Fig. 9; Figs. A4, A5; Supplementary
Tables 25, 26).

For detrended mean annual time scales (July-June) and austral summer (JFM) the

Cabri SST record shows a positive correlation with southern Indian Ocean SST along a
southeast to northwest band stretching along the trade wind belt (Figure 9d-f; Fig. A4).
The correlation with the southern Indian Ocean trade wind belt remains stable over
different record length and is most pronounced post 1971. We also find positive
correlations with the Bay of Bengal and the Maritime Continent throughout the past 60
years. We find positive correlations with the eastern Pacific SST and negative
correlations with the northern Pacific along 40°N and stretching between 160°E and
150°W. The SST pattern mimics part of the typical spatial ENSO and PDO pattern across
the Indo-Pacific (Mantua et al., 1997; McPhaden et al., 2006).

**6 Discussion**
**6.1 Diagenesis, orientation of corallites and potential biases in Sr/Ca derived SST**

Diagenesis could be excluded as a major cause of discrepancies between coral

SST and grid-SST. For core Totor, only for the period between 1882 and 1887 we have to
assume that diagenesis caused a cool bias on our coral SST reconstruction (Figure 8).



Core Cabri showed only localized positive Sr/Ca anomalies (cool SST bias) most
probably caused by aragonitic sediment trapped within growth framework pores. These
samples have been removed before interpolation. Having excluded diagenesis for most of
the record, we assessed sampling biases due to changes in the orientation of growth axes
and positioning of corallites to the slab surface. De Long et al. (2012) showed clear
evidence for warm or cool biases in coral Sr/Ca-SST reconstructions caused by
suboptimal orientation of corallites in corals from New Caledonia. We have adopted a
similar approach to test for sampling biases in our two cores (Table 2 & 3). We found
that core Totor contained areas where a sampling bias could explain anomalous Sr/Ca-
derived SST. The warm anomaly between 1916 and 1921 with its peak values in 1919
stands out as the largest single anomaly in the record. However, diagenesis cannot
explain the warm anomalies. The growth rates and Sr/Ca seasonality for all years
between 1916 and 1921 are not anomalous and close to the average seasonality from *in*
*situ* SST data. The orientation of the corallites is mostly optimal (parallel to slab surface)
to the surface. However, for the years 1916 to 1921 we recognized an interval with
bundles of oblong corallites where our sampling transects switched from optimal to
suboptimal growth orientation. De Long et al. (2012) showed that warm biases were often
caused by corallites orientated at an angle or oblong to the slab surface and where growth
orientation had changed. These suboptimal intervals have seasonal cycles with more
summer Sr/Ca values than winter values causing an apparent warm bias. The latter could
not be identified for core Totor 1918-1919 values. Nevertheless, the extreme warm
anomaly between 1916 to 1921 is most likely related to the change in growth direction
associated with an unidentified vital effect. Interestingly, despite the potential influence



of vital effects on the trend, the seasonality in this core section was well preserved. This
implies that seasonality can be captured robustly while absolute values and trends are
potentially biased by vital effects. This adds confidence for the study of seasonality from
fossil corals where vital effects are harder to distinguish from true variability due to the
lack of SST data for verification.

The warm anomalies in the periods 1854-1860, 1936-1941 and 1948-1951 in core

Totor are all associated with an orientation of corallites at an angle to the slab surface.
Yet, the interval 1936 to 1941 shows a high growth rate and normal seasonality in Sr/Ca
for all years and no extreme over-representation of summer versus winter samples. The
intervals 1948 to 1951 and 1854 to 1860 both showed reduced growth and seasonality
which might have caused apparent warmer winter Sr/Ca values. We also detected areas
with warm anomalies for years that predate instrumental data coverage (Tab. 2). The
1820's and 1830's likely have a warm bias due to corallites at an angle, disorganized fans
and reduced growth rate with more summer values (Tab. 2). Between 1798 and 1816, the
orientation is optimal and no bias can be inferred. The years pre-1798 have to be
considered with caution since the bottom of the core Totor did show disorganized fans at
places and/or suboptimal orientations pointing to likely warm biases (indicated in Figure

6).

Between 1984 and 2005 (core tops), Sr/Ca trends in cores Totor and Cabri deviate

with Totor showing a cooling trend while Cabri shows a strong warming trend (Fig. 7).
Our analysis of growth orientation revealed that the corallites in core Totor form parallel,
elongated rods of septa for the entire period 1984 to 2005 (Fig. 7) while Cabri does show
an optimal orientation of corallites for the core top between 1984 and 2006 (Fig. 7), with



the exception of sub-optimal corallites in the period 2000 to 2006. The peculiar structure
of the corallites in Totor, at a first glance, would suggest optimal vertical growth of the
corallites with the polyps clearly visible from the apex of the core slab. However, this
structure is clearly associated with high Sr/Ca ratios and articifially cold SST anomalies.
A similar structure of the corallites was found in *Porites lutea* from St. Marie Island off
East Madagascar (core STM4 in Grove et al. 2013). Grove et al. (2013) ascribed the
Sr/Ca trend difference between cores STM2 and STM4 to changes in coral growth and
calcification, yet their results were not conclusive. Re-examination of core STM4
revealed that it also forms the parallel-elongated rods of septa in the core top, which was
biased towards high Sr/Ca ratios and therefore cold SST anomalies. STM4 also showed
low densities in this core top section that agrees with low density in Totor. Inspection of
various core sections in Totor and other coral cores revealed that similar elongated rods
of septa (not sampled down core) are formed between neighboring growth fans of septa.
We suggest that these parallel septa grow very fast in summer and winter, therefore show
no clear density contrast with overall low skeletal density. The Sr/Ca seasonality is also
strongly enhanced and samples contain a higher number of winter samples that record
high Sr/Ca ratios in Totor. Interestingly, the summer Sr/Ca values between cores Totor
and Cabri agree rather well between 1984 and 2005 while the winter values in Totor are
strongly biased to extreme cold anomalies. We suggest that core tops from *Porites* sp.
with similar parallel septa should be avoided for sampling since it can cause a cold bias in
Sr/Ca-based SST reconstructions.

Overall, our test for sampling biases to a large extend confirms the findings of De

Long et al. (2012) and indicates that such analysis should accompany climate



reconstructions from coral cores. Our results suggest that a new core needs to be obtained
from the Totor colony or other large *Porites* sp. in order to overcome the SST biases
identified in the current record. The Cabri coral (>3.5m in height) would be an ideal site
since for the period 1945 to 2006 it provided an excellent and largely un-biased record of
SST. Yet, the 1907 dead surface was present in three long cores drilled from the Cabri
coral at different angles, which could undermine the SST reconstruction for a few
decades below the mortality event. The reason for the mortality event could not be
determined.

**6.2 SST trends and large-scale climate teleconnections since 1945**
Based on our analysis of corallite orientations, we conclude that core Cabri is
most likely the best representation to assess SST trends and interannual variability since
1945. Nevertheless, trend estimates in both individual cores and for the composite record
need to be interpreted with caution (as indicated in Figure 6).
Both, the Cabri and coral composite time series show an increase in SST over the
past 60 years (since 1945; Figs. 5 and 6). The Cabri time series recorded a higher SST
rise (1.38±0.41°C) than instrumental data, which ranged between 0.61 to 0.86±0.15°C,
and the composite coral record (0.44±0.37). The trend in Cabri agrees with all SST
datasets within 2σ, whereby the lower range of uncertainty for the Cabri trend estimates
(~1°C) and the upper range for the coral composite (~0.8°C) is in closes agreements to
trends from gridded SST datasets. Most of the accelerated warming trend in Cabri
resulted from the recent 6 years where the orientation of the corallites was sub-optimal.
The composite record agrees with the trend in AT at Rodrigues and marine AT



(HadMAT1, HadNMAT2) within 1σ, yet likely underestimates the trend in grid-SST
(Fig. 5; Figs. A5, A6). The AT record shows very warm anomalies for the years 1951 to
1955 which resulted in a lower long-term trend. The composite record also showed warm
years between 1951 and 1955 due to core Totor that indicated warm SST while Cabri
followed grid SST with colder temperatures (Fig. 5). The Totor site is a sheltered location
with light winds and restricted water movement, with all three having contributed to
severe bleaching in 2002 at this site (Hardman et al., 2004, 2008). It could well be that
core Totor has at times recorded local SST variations that do not reflect open ocean
conditions or those at the more exposed site Cabri. This site-specific, local SST
variability might partly explain the high SST anomalies in Totor between 1936 and 1941
where the orientation of the corallites did not conclusively accounted for Sr/Ca-SST
anomalies. We conclude that the SST trend in Cabri and the coral composite closely
follows open ocean grid-SST which both indicate strong warming (~0.68-1°C) of the
south-central Indian Ocean over the past 60 years. Roxy et al. (2014) reported that during
1901–2012, the Indian Ocean warm pool warmed by 0.78°C while the western Indian
Ocean (5°S–10°N, 50°–65°E) experienced anomalous warming of 1.28°C in summer
SSTs. Our results for Cabri are therefore not unusual and within the range of observed
Indian Ocean SST trends (Annamalei et al., 2005; Alory et al., 2007; Koll Roxy et al.,
2014). The strong warming in the southern Indian Ocean trade wind belt could potentially
alter the monsoon circulation, especially during the monsoon onset phase in austral
autumn (March to May; Annamalei et al., 2005). Both, our coral SST time series and SST
products indicate the strongest warming for the March to May season (not shown).
Rodrigues station precipitation is strongly positively correlated with SST between March





and May. When precipitation is anchored over a warmer SWIO between March and May
it can prevent the movements of the ITCZ towards the North and potentially disrupt the
Asian monsoon onset (Annamalei et al., 2005).

Both the Cabri and coral composite SST reconstructions revealed a clear

ENSO/PDO teleconnection pattern for mean annual and austral summer averages with
positive correlations across the Indian Ocean resembling the Indian Ocean basin mode
pattern (Xie et al., 2016) in response to ENSO and PDO (Fig. 9). Cabri shows the
strongest teleconnection pattern, which suggests that this time series is the most reliable
between 1945 and 2006 to assess ENSO/PDO impacts on Rodrigues (Figs. A3 to A5).
The ENSO/PDO teleconnection was stable for the recent 60 years, yet was strongest
between 1971 and 2006 (Fig. 9c,f). The latter period is known for increased occurrence of
El Niño events and a switch to a positive PDO phase up to 1999 (McPhaden et al., 2006).
These results are in agreement with ENSO/PDO pattern correlations observed in other
coral records from the southwestern Indian Ocean (Pfeiffer et al., 2004; Crueger et al.,
2009). However, this is the first Indian Ocean coral SST reconstruction that shows a clear
relationship with the PDO, while other coral records reflected PDO relationships with
rainfall/river runoff (Grove et al., 2013) and salinity/sea level pressure (Crueger et al.,
2009; Pfeiffer et al., 204).

Coral reefs of Rodrigues escaped the mass coral bleaching event of the 1997–

1998 El Niño, yet experienced bleaching in February 2002, March-April 2005 and April-
May 2006 (Hardman et al., 2004, 2008). The most severely affected sites with highest
coral mortality were located in the north and west of the island with our site Totor located
within the zone of most severely affected reefs in 2002, 2005 and 2006 (Hardman et al.,



2004, 2008). Our site Cabri showed only 11-30% bleached corals in 2005, yet less severe
impacts in 2006 and 2007 and appears less frequently impacted by anomalously high SST
during recent El Niño events. Hardman et al. (2008) concluded that coral bleaching at
Rodrigues is very patchy and to date most sites appear to be resilient to current El Niño
thermal stress events. The relatively large seasonal SST amplitude (6.22°C) and high
standard deviation (2.14°C) might serve as buffer to prevent extended periods of thermal
stress events during El Niño events. Degree heating weeks for Rodrigues post 1998 rarely
exceeded 4 weeks and only in 2002 and 2005 reached 8 weeks at the northern and north-
western coral reef sites which have experienced severe thermal stress and are in decline
(Hardman et al., 2008). Despite the strong warming trend and El Niño related thermal
stress observed in our study, the corals of Rodrigues appear to be a safe haven for coral
survival. However, expected levels of future warming in the coming decades will
increase thermal stress levels and probably increase coral bleaching and mortality.
Rodrigues receives a very limited larval supply suggesting that the reefs rely on larval
retention and self-seeding for population recovery. Gilmour et al. (2013) and Graham et
al. (2015) showed that isolated reefs with limited larval supply might be the more
susceptible to climate change-driven reef degradation, despite escaping many of the
stressors impacting continental reef systems. It is therefore most important to reduce local
stressors at Rodrigues to provide the corals enough time to bounce back after thermal
stress disturbance.

**7 Conclusions**



We reconstruct SST for Rodrigues Island located in the south-central Indian Ocean trade
wind belt. Our reconstruction is based on two monthly-resolved coral Sr/Ca records
(Totor, Cabri) from Rodrigues Island (63°E, 19°S) that extend to 1781 and 1945,
respectively. We identify potential biases in our SST reconstructions associated with the
orientation of the corallites and conclude that careful screening for diagenesis and
orientation of corallites is of paramount importance to ensure high quality of Sr/Ca-based
SST reconstructions. However, our proxy records provide the most reliable SST
reconstruction between 1945 and 2006 and for several multi-decadal periods over the past
224 years. Reconstructed long-term SST trends are within the range of trends reported
from observational SST data for the western Indian Ocean. Furthermore, we identify
teleconnections with the ENSO/PDO over the past 60 years, eg. warming of SST during
El Niño or positive PDO. Our reconstruction is the first coral proxy record for SST that
shows a relationship with the PDO spatial correlation pattern in SST. We suggest that
Rodrigues Island is an ideal site to assess SST variations in the southern Indian Ocean
trade wind belt and their climatic teleconnection with the ENSO/PDO on longer time
scales.

**8 Acknowledgements**

The coral paleoclimate work was supported as part of the SINDOCOM grant

under the Dutch NWO program 'Climate Variability', grant 854.00034/035. Additional
support comes from the NWO ALW project CLIMATCH, grant 820.01.009, and the
Western Indian Ocean Marine Science Association through the Marine Science for
Management program under grant MASMA/CC/2010/02. We thank the team of



SHOALS Rodrigues for their excellent support in fieldwork logistics and in the
organization of the research and CITES permits. We would also like to thank the
Rodrigues Assembly and the Ministry for Fisheries for granting the research and CITES
permits. A Senior Curtin Fellowship in Western Australia, and an Honorary Fellowship
with the University of the Witwatersrand, South Africa, supported JZ. Bouke Lacet and
Wynanda Koot (VUA) helped cut the core slabs and prepared the thin sections. Janice
Lough and Eric Matson (AIMS) provided skilled technical support for coral core
densitometry measurements and data processing. We thank Dieter Garbe-Schoenberg for
assistance with the ICP-OES measurements.

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

**Appendix A – Coral growth data and comparison to instrumental temperature**
**records**

**Tables**

| Core name | GPS position | Species | Water depth (m) | Mean growth rate cm year$^{-1}$ | Mean density g/cm$^3$ | Mean Calcification rate g/cm$^2$ year$^{-1}$ |
|---|---|---|---|---|---|---|
| **Totor** | S19º40.237; E63º25.754 | *Porites sp.* | 4.0 | 0.92 (±0.19) | 1.128 (±0.11) | 1.07 (±0.18) |
| **Cabri** | S19°40.030, E63°26.065 | *Porites lobata* | 3.0 | 1.18 (±0.25) | 1.36 (±0.12) | 1.60 (±0.16) |

Table 1 - Coral cores with their GPS co-ordinates and depths at low tide, with mean rates
of extension, densities and calcification over the complete length of the individual
records.











| Section | Year | Orientation | Bias | Notes |
|---|---|---|---|---|
| 1 | 2005-1987 | Sub-optimal | cool | Corallites parallel to surface, yet straight angle; probably like a valley |
| 2 | 1987-1982 | Sub-optimal | cool | Corallites parallel to surface, yet straight angle; probably like a valley |
| 2 | 1981-1977 | Sub-optimal | warm | Corallites at an angle to the surface; oblong corallites |
| 3 | 1978-1975 | Sub-optimal | warm | Corallites at an angle to the surface |
| **3** | **1974-1958** | **Optimal** | **none** | **Corallites parallel to surface** |
| 4A | 1958-1952 | Sub-optimal | warm | Corallites at an angle to the surface; scallop texture from angles of corallites |
| 4A | 1951-1945 | Sub-optimal | warm | Corallites at an angle to the surface; 1947-1952 low growth rate; reduced seasonality |
| **4B** | **1947-1936** | **Optimal** | **none** | **Corallites parallel to surface, 1945-1947 better orientation than in slab 4A** |
| **4B** | **1938-1933** | **Sub-optimal** | **none** | **Corallites at an angle to the surface; 1936-1941 warm anomaly years show normal seasonality and high growth rate** |
| **5** | **1933-1922** | **Optimal** | **none** | **Corallites parallel to surface; 1922-1928 reduced seasonality** |
| 6 | 1921-1915 | Sub-optimal | warm | 1915-21 warm spikes shows slightly oblong corallites, yet normal seasonality; switch from optimal to sub-optimal orientation |
| **6** | **1915-1896** | **Optimal to sub-optimal** | **none** | **Corallites mostly parallel to surface, small section with corallites at slight angle;;** |
| **7** | **1897-1890** | **Optimal** | **none** | **Corallites parallel to surface** |
| 7 | 1887-1882 | Optimal | cool | Diagenesis detected between years 1882-1887 |
| **7** | **1881-1872** | **Sub-optimal** | **none** | **Corallites at an angle to the surface; 1872 close to bioerosion track; 1878-1880 low seasonality, yet no effect** |
| 8 | 1872-1868 | Sub-optimal | cool | Corallites at an angle to the surface; some corallites at almost 90° angle; 1868-1872 below bioerosion track; 1867-1871 low seasonality |
| 9 | 1860-1854 | Sub-optimal | warm | Corallites at an angle to the surface; 1854-1858 low seasonality, less winter samples |
| 9 | 1856-1845 | Sub-optimal | warm | Corallites parallel to surface; low seasonality with relatively warm winter samples |
| **9** | **1844-1831** | **Optimal** | **none** | **Corallites parallel to surface; only 1831-1832 corallites at an angle to surface** |
| 10 | 1830-1827 | Sub-optimal | warm | Corallites at an angle to the surface; oblong orientation |
| 10 | 1826-1823 | Disorganised | warm | Corallites rotating at 90° angle; low growth rate, seasonality reduced 1823-1825 with relatively warm winter samples |
| **10** | **1822-1815** | **Optimal** | **none** | **Corallites parallel to surface; low growth rate; reduced seasonality 1818-1822, yet no effect on SST anomalies** |
| **11** | **1816-1806** | **Sub-optimal** | **none** | **Corallites at an angle to the surface, yet no effect on SST anomalies** |
| **11** | **1807-1798** | **Sub-optimal** | **none** | **Corallites at an angle to the surface in sub-optimal parts; Corallites rotating at 90° angle near terminating fans (not sampled); 3 growth axes with terminating fans in between (not sampled); 1799-1807 regular seasonality** |
| 11 | 1797-1792 | Sub-optimal | warm | Corallites at an angle to the surface |
| 12 | 1795-1792 | Disorganised | warm | Corallites rotating at 90° angle; 1792-1791 long year, more summer samples |
| 12 | 1791-1784 | Sub-optimal | warm | Corallites parallel to surface; 1784-1787 Corallites at an angle to the surface; 1789-1794 seasonality distorted |
| 12 | 1781-1783 | Disorganised | warm | Corallites rotating at 90° angle; seasonality slightly distorted, apparently more summer samples |

Table 2 – Summary of sampling issues detected in core Totor. Unbiased sampling tracks
indicated in bold.






| Section | Year | Orientation | Bias | Notes |
|---|---|---|---|---|
| 1 | 2007-2000 | Sub-Optimal | warm | Corallites parallel to surface; yet no clear growth fans |
| **1** | **1999-1992** | **Optimal** | **none** | **Corallites parallel to surface** |
| **2** | **1984-1992** | **Sub-optimal** | **none** | **Corallites at an angle to the surface; oblong corallites** |
| **3** | **1983-1968** | **Sub-Optimal** | **none** | **Corallites parallel to surface; yet no clear growth fan** |
| **4** | **1967-1964** | **Sub-optimal** | **none** | **Corallites at an angle to the surface** |
| **5** | **1963-1958** | **Optimal** | **none** | **Corallites parallel to surface** |
| **5** | **1957-1954** | **Sub-optimal** | **none** | **Corallites at an angle to the surface** |
| **5** | **1953-1945** | **Optimal** | **none** | **Corallites parallel to surface** |


Table 3 – Summary of sampling issues detected in core Cabri. Unbiased sampling tracks
indicated in bold.

**Figure captions**
Figure 1 – Map of Rodrigues island with the position of the two corals cores at Totor and
Cabri indicated. The star shows the position of the CTD that collects SST and salinity
data. Polygon indicates the location of the Meteorological Station which records air
temperature, sunshine hours, wind speed and rainfall.
Figure 2 – Climatology at Rodrigues between 1997 to 2007. A) SST *in situ*, ERSSTv.3
(Smith et al., 2008) and AVHRR SST from NOAA Coral Reef Watch (Reynolds et al.,
2007); b) air temperature and sunshine hours at Rodrigues Meteorological Station (MET);
c) monthly averaged wind speed at MET.

Figure 3 – a) Time series of monthly (thin solid lines) Sr/Ca anomalies (right Y-axis)
relative to the 1961 to 1990 climatological mean for coral cores Cabri (top), Totor
(middle) and Coral composite (bottom) for the period 1781 to 2006.



896 Figure 4 – Reconstructed absolute SST from coral Sr/Ca from cores Totor and Cabri for

897 2002 to 2006 based on calibration with in situ SST from Rodrigues. The uncertainty for

898 single month absolute SST for individual cores Cabri and Totor is 1.23°C and 1.05°C

899 (1σ), respectively. The coral data agree with *in situ* SST within the 1σ uncertainty.

900

901 Figure 5 – Time series of annual mean temperatures anomalies relative to the 1961-1990

902 mean for the coral composite SST, Rodrigues weather station Air temperature (AT),

903 ERSSTv3b, ERSSTv4 , HadISST, HadSST3, HadMAT1 and HadNMAT2 for the period

904 1950 to 2006. The uncertainty of mean annual coral Sr/Ca-SST anomalies are indicated

905 by the grey envelope.

906

907 Figure 6 – Annual mean time series of coral time series (red) for a) Cabri, b) Totor and c)

908 the coral composite SST compared to SST reconstructions: ERSSTv3b, ERSSTv4,

909 HadISST, HadSST3, HadMAT1 and HadNMAT2. See legend in a) for colours. For all

910 time series we computed anomalies relative to 1961 to 1990. The uncertainty of mean

911 annual coral Sr/Ca-SST anomalies are indicated by the grey envelope. Potential warm

912 bias in coral SST is indicated by faint red shading, while cool bias by light blue shading.

913 Yellow marks core intervals with diagenesis.

914

915 Figure 7 – a) Monthly interpolated Sr/Ca profiles for cores Cabri (red) and Totor (grey).

916 B) Images of core Totor (coloured blue) with orientation of corallites indicated. Years for

917 core sections indicated on coral slab and grey arrow points to major change in orientation

918 of corallites in core top section of Totor around 1983/84.






Fig. 8: Thin-section and scanning electron microscope (SEM) images. Thin section
photographs are shown in plane- (left) and cross-polarized light (middle). A and B:
Excellent preservation of coral skeleton without dissolution or cementation is typical for
the corals Totor and Cabri. Small patches of aragonite cements occur in parts of slab 6
(C), 7 (D) and 11 (E) of Totor. F (left): A prominent growth break visible in the
radiograph of slab 12 of Totor is characterized by abundant microborings and
encrustation by coralline red algae. F (middle): The section above the growth break is
well preserved. F (right): The coral core Cabri shows excellent preservation, only locally
containing aragonitic sediment partially filling pore spaces.

Figure 9 – Spatial correlation of Cabri Sr/Ca-SST anomalies (relative to 1961-1990) with
HadISST (Rayner et al., 2003). January to March austral summer in a) between 1945-
2006, b) 1961-1990 and c) 1971-2006. Annual mean correlations in d) between 1945-
2006, e) 1961-1990 and f) 1971-2006. Only correlation with $p<0.05$ are coloured.
Computed at knmi climate explorer (van Oldenborgh and Burgers, 2005).

Figure A1 – Relative changes in coral growth parameters (anomalies relative to 1961-
1990) of cores Totor (dark grey; since 1836) and Cabri (light grey; since 1907) versus
Rodrigues coral composite SST (black solid line) for period of best geochemical data
coverage.

Figure A2 –Number of SST observations in the grid box surrounding Rodrigues in the
ICOADS database. Note the extremely sparse observations even in recent years (van
Oldenborgh and Burgers, 2005).



Figure A3 – Spatial correlations of global austral summer HadISST for 30-year periods
with a-f) austral summer coral composite summer SST (January to March) for different
30-year periods. Only correlations with p<0.05 coloured. Computed at knmi climate
explorer (van Oldenborgh and Burgers, 2005).

Figure A4 – Spatial correlations of Cabri coral SST with global austral summer HadISST
for a-c) 1950 to 1975 (February to May) negative PDO phase (Mantua et al., 1997) and c-
d) 1976 to 1999 (January to April ) positive PDO phase. Only correlations with p<0.05
coloured. Computed at knmi climate explorer (van Oldenborgh and Burgers, 2005).
Figure A5 – Spatial correlations of mean annual HadMAT1 air temperature anomalies
between 1945 to 2001 relative to 1961-1990 with a) HadISST for Rodrigues, b) coral
composite SST and c) Cabri SST. Only correlations with p<0.05 coloured. Computed at
knmi climate explorer (van Oldenborgh and Burgers, 2005). Y-axis Latitude, X-axis
Longitude.

Figure A6 – Coral composite monthly SST anomalies relative to 1961-1990 (red)
compared to 5°x5° gridded HadNMAT2 night marine air temperature (blue; Kent et al.,
2013). The uncertainty of coral SST based on the regression slope error is indicated by
the grey envelope. Note the excellent agreement between the monthly anomalies.
Summer (Dec-April) and Winter (June-August) anomalies are correlated with r=0.5,
p<0.001 (N=56).





Figure A7 – X-ray positive print for slabs of core Totor with sampling lines indicated.
Blue lines indicate high resolution sampling tracks. Yellow lines superimposed on blue
lines indicate sampling at annual resolution for other purposes. Start or end years for each
slab indicated.

Figure A8 - X-ray positive print for slabs of core Cabri with sampling lines (milling
holes) indicated. Start or end years for each slab indicated. Note the dead surface before
1907 that is most probably related to a past coral bleaching event.

Table A1 – Statistics of various sea surface temperature (SST) products and air
temperature for Rodrigues with 1σ standard deviations in brackets for the period 2002 to
2006 (period with *in situ* SST data). STDV = 1σ standard deviation over all years. All
units in °C.

Table A2 - Linear regression of coral Sr/Ca with a) *in situ* SST 2002-2005/6, b)
ERSSTv.3 (Smith et al., 2008) 1997-2005/6, c) AVHRR SST NOAA Coral Reef watch
data 2000-2005/6 and d) monthly Sr/Ca with AVHRR SST (Reynolds et al., 2007) for the
period 1982 to 2005.









**Figures**

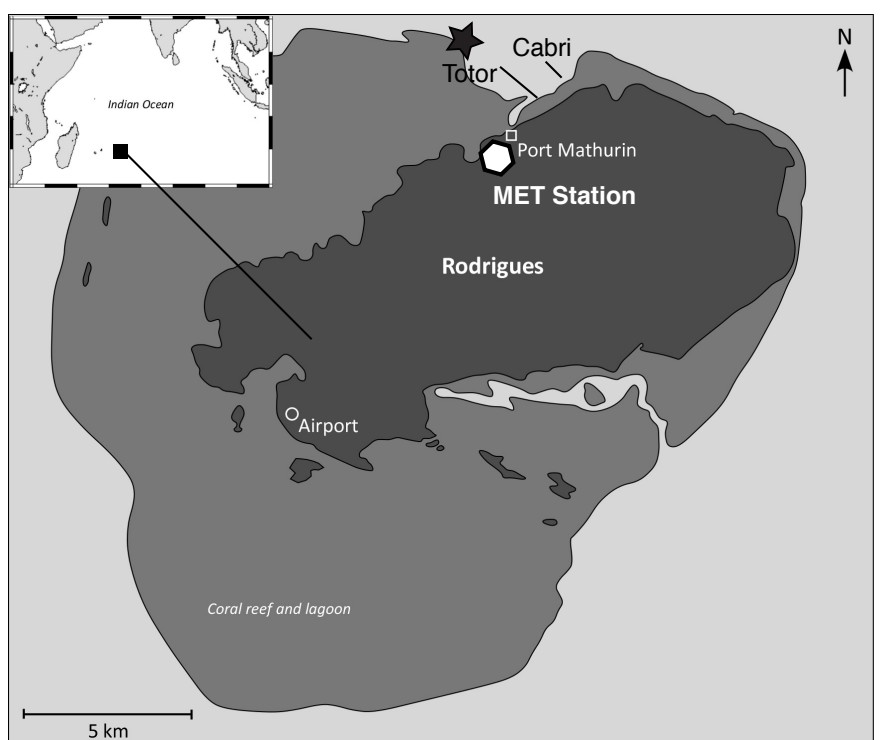


Figure 1





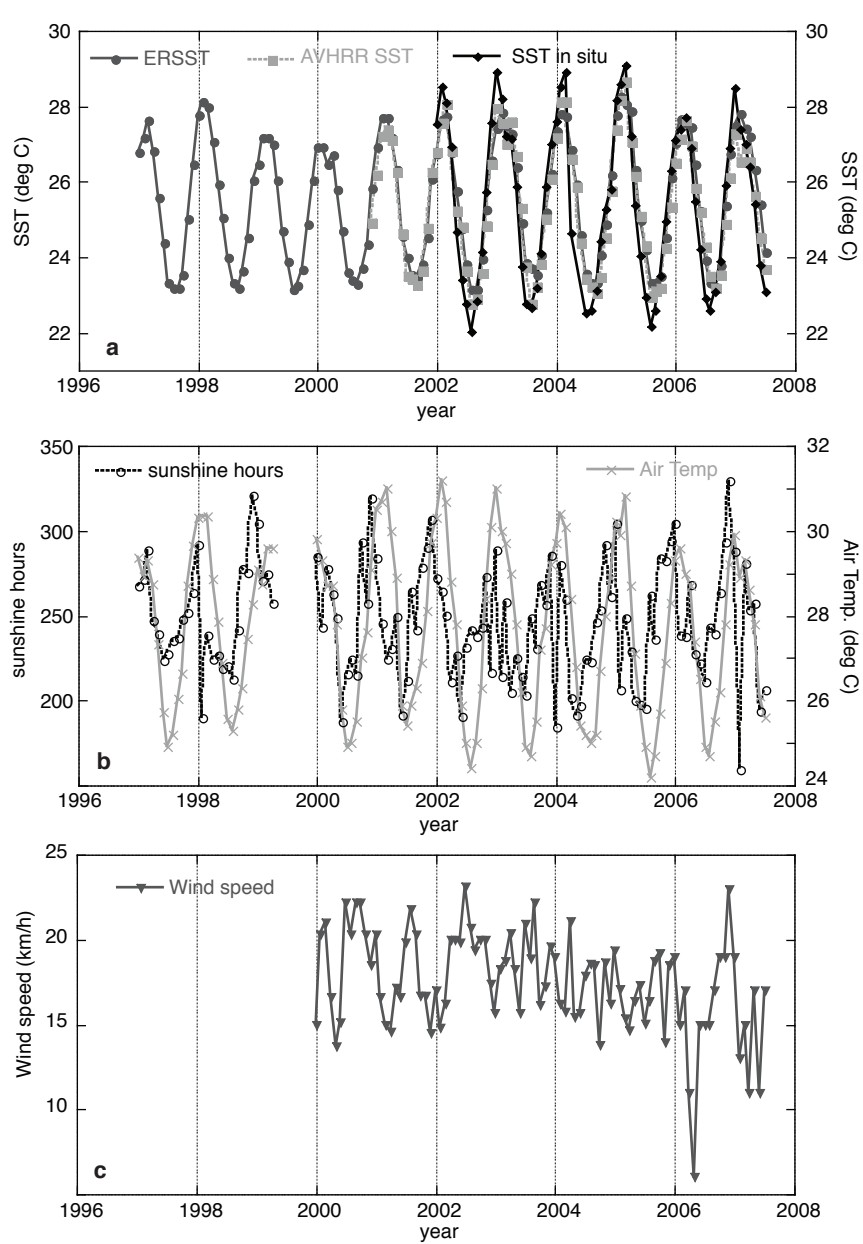

**Figure 2**


Figure 2



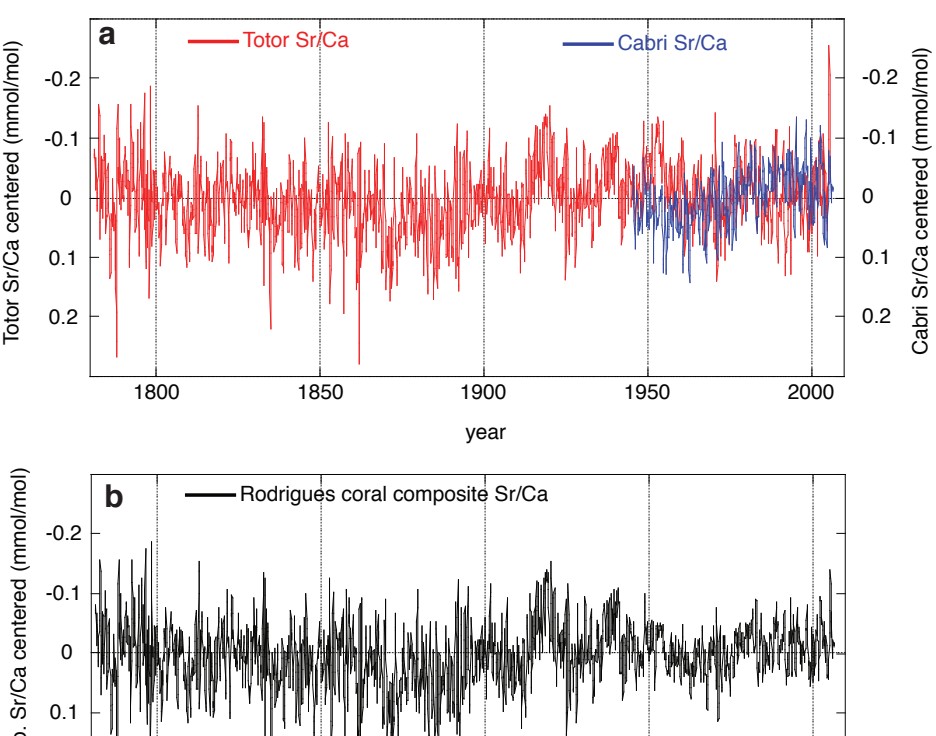


Figure 3











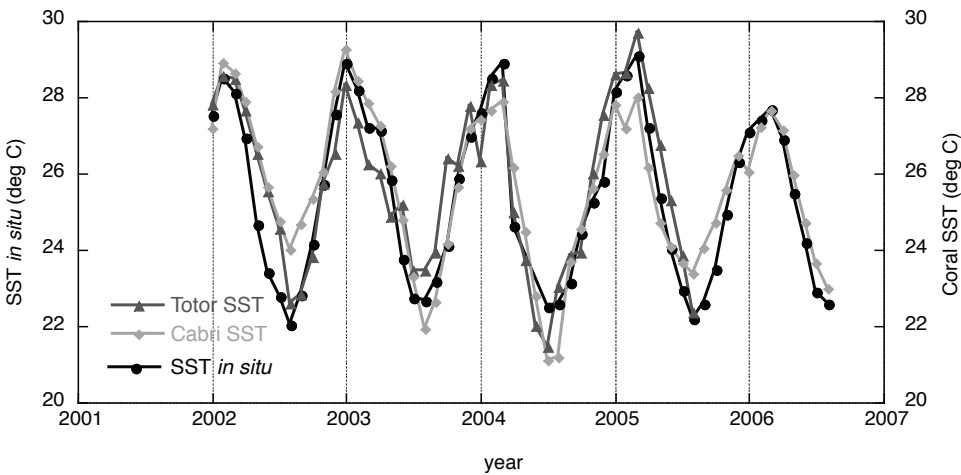


Figure 4

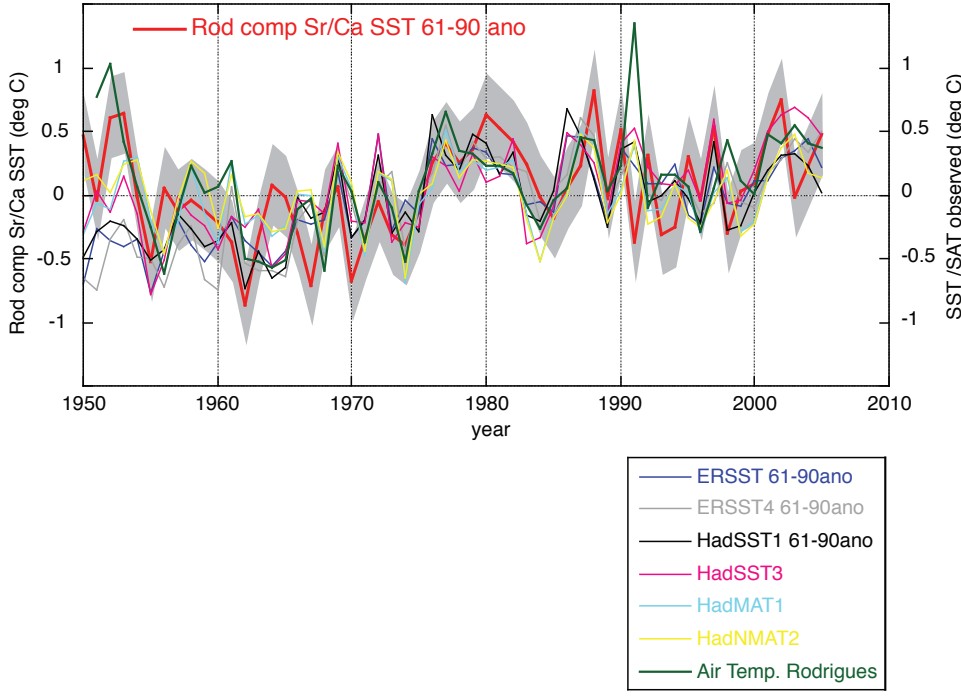


Figure 5



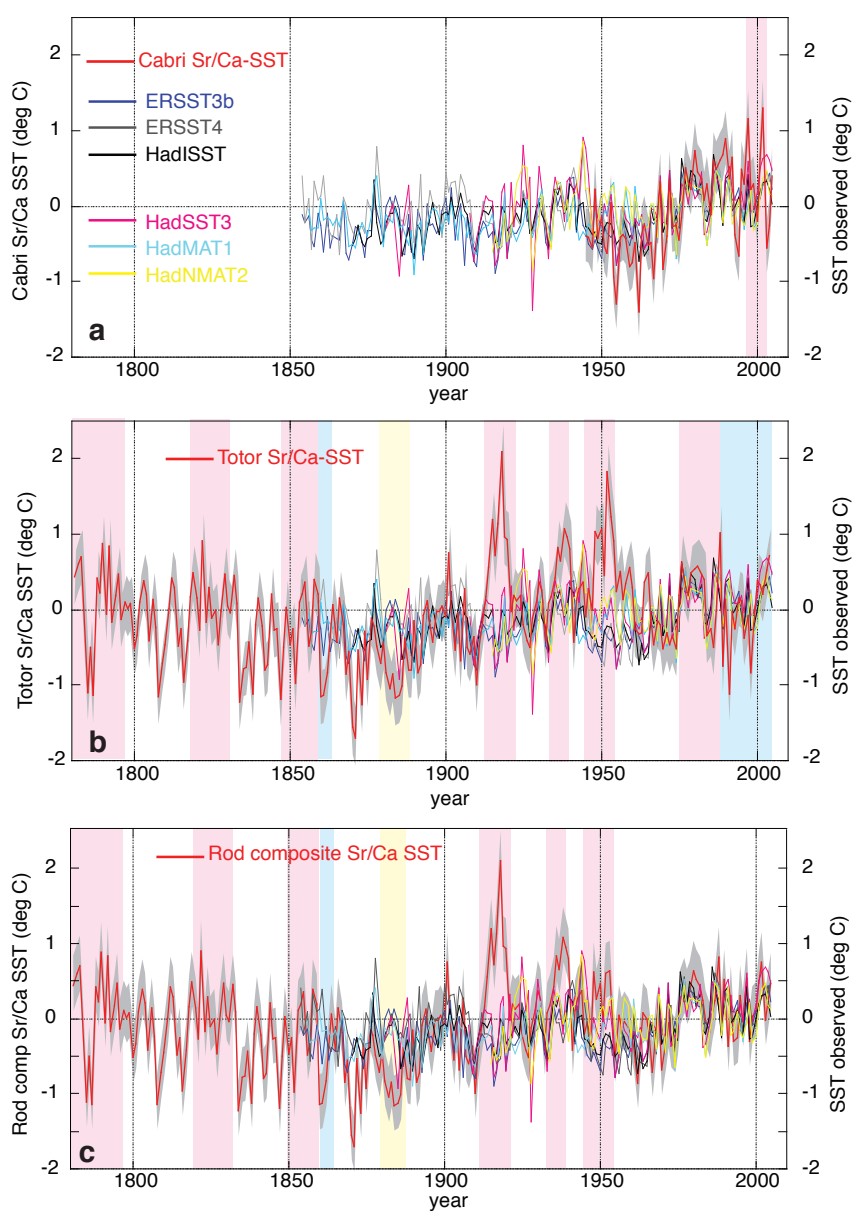


Figure 6





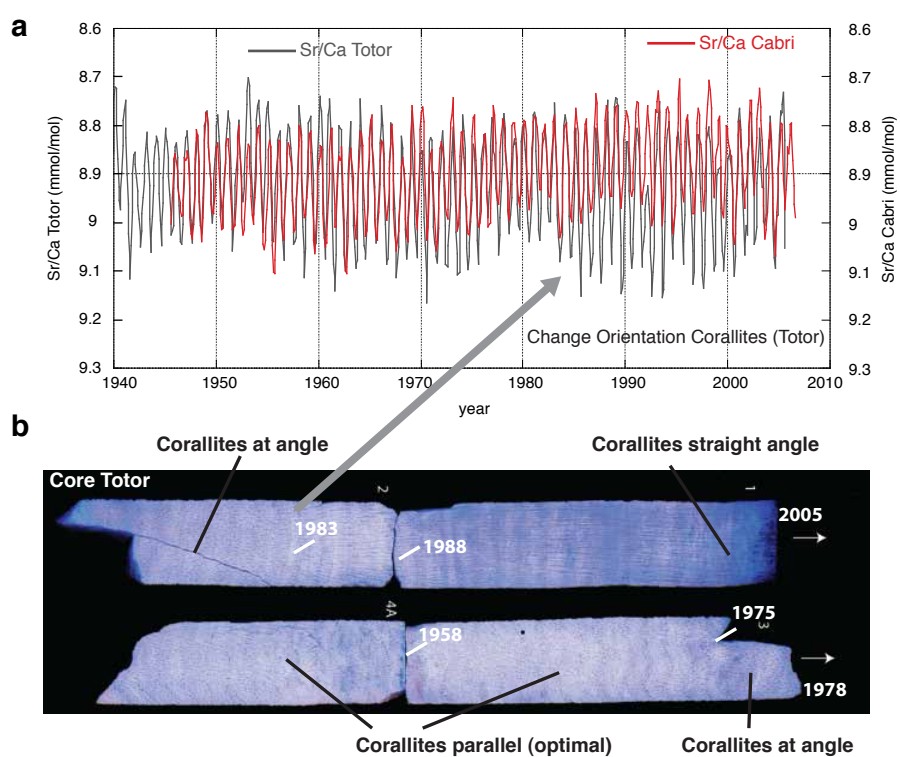


Figure 7



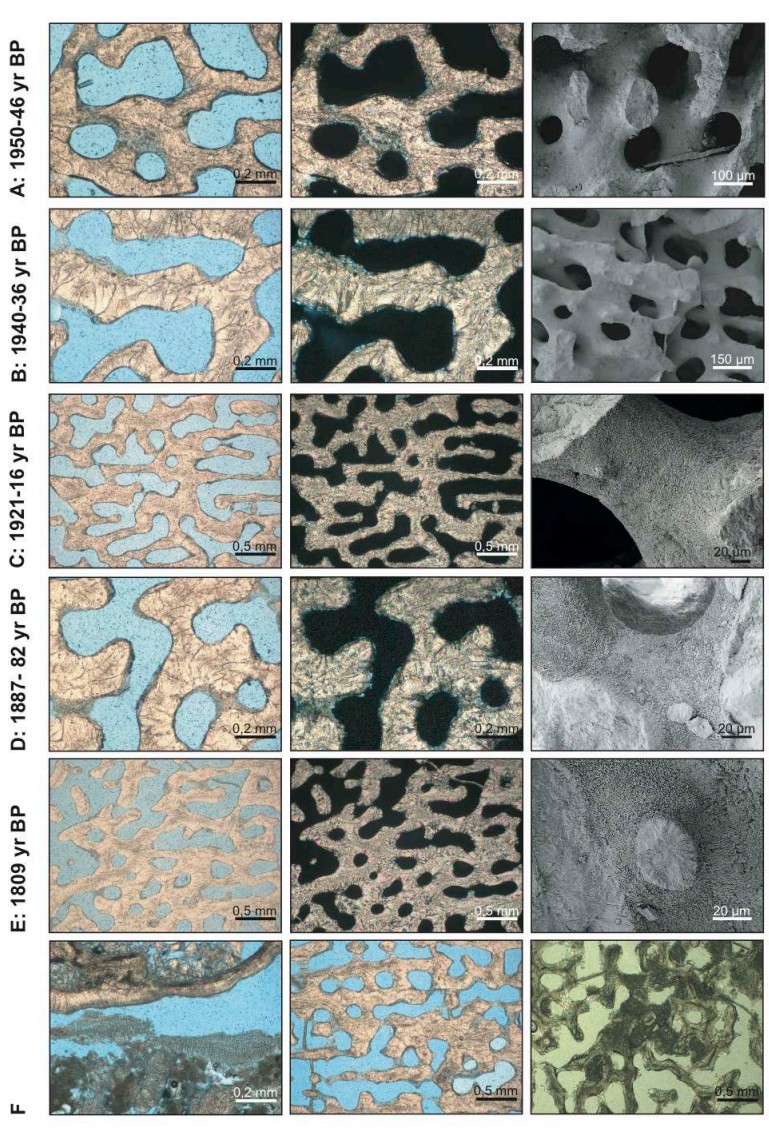

Fig. 8


Figure 8





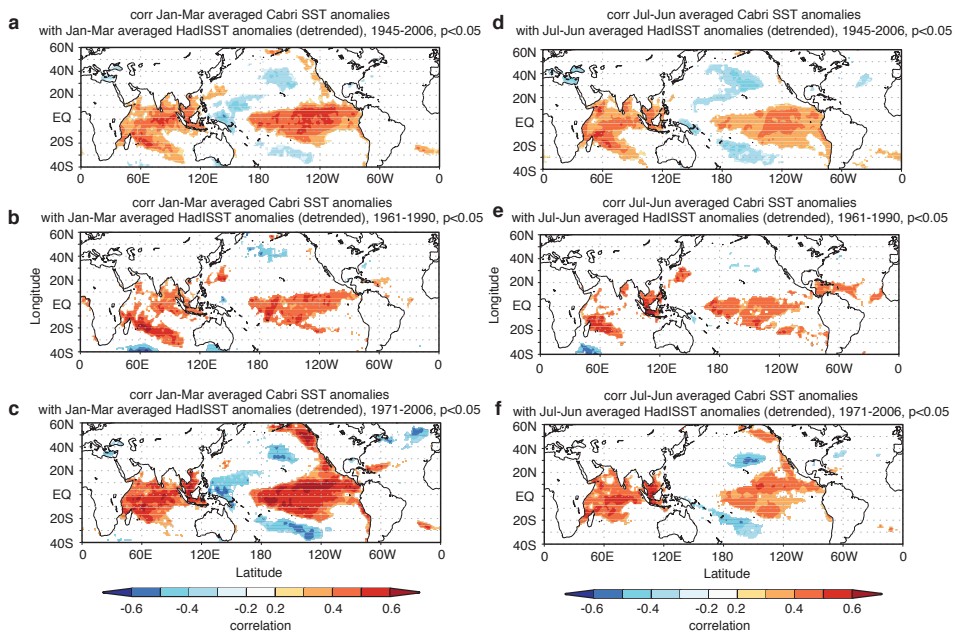


Figure 9





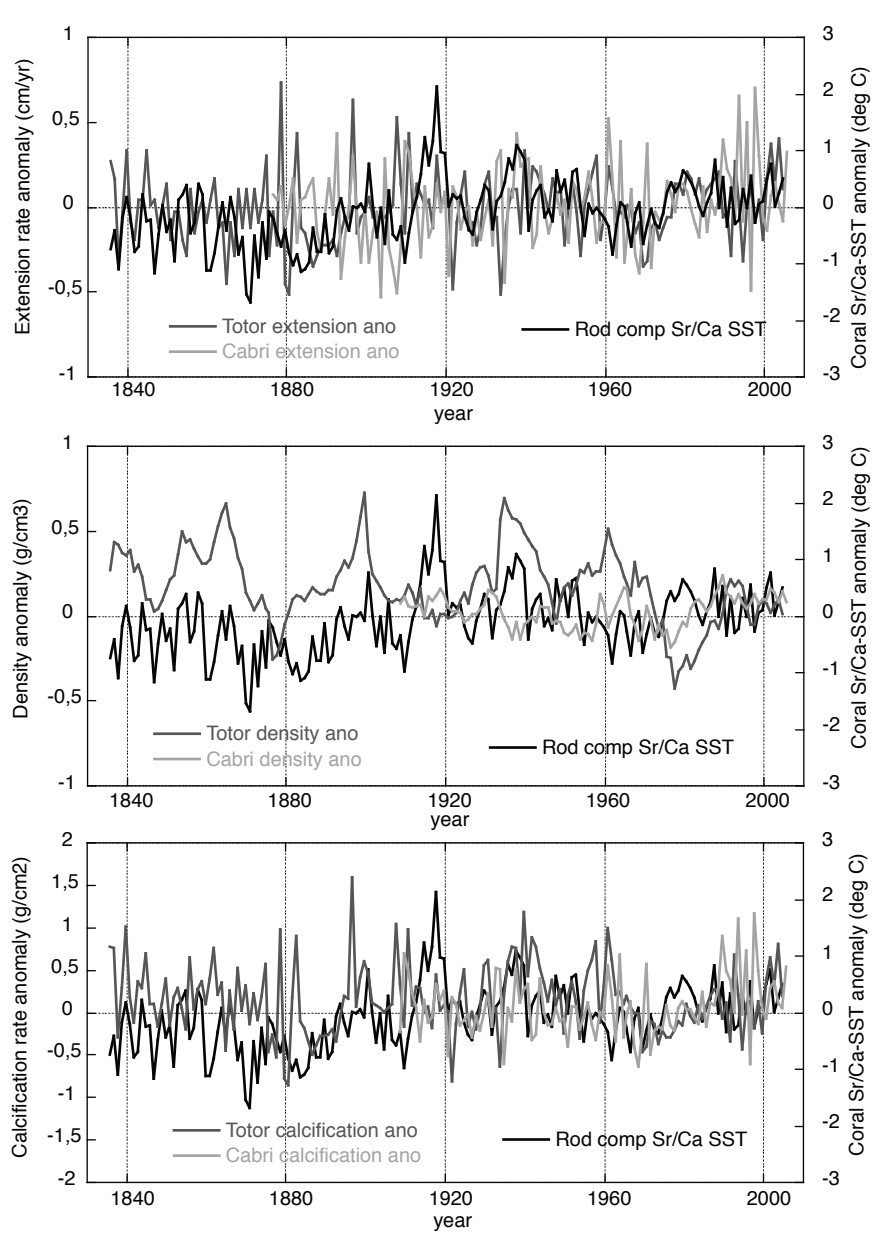


Figure A1





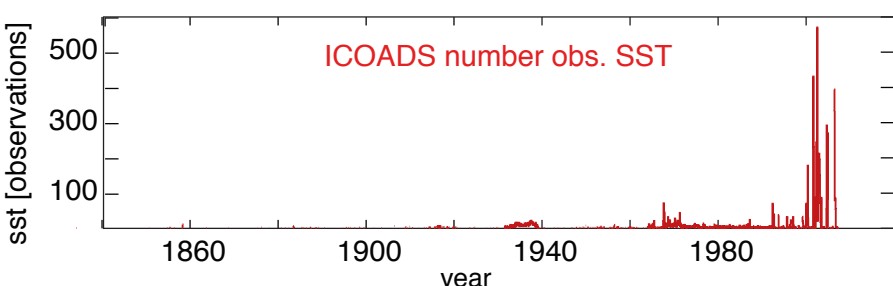


Figure A2

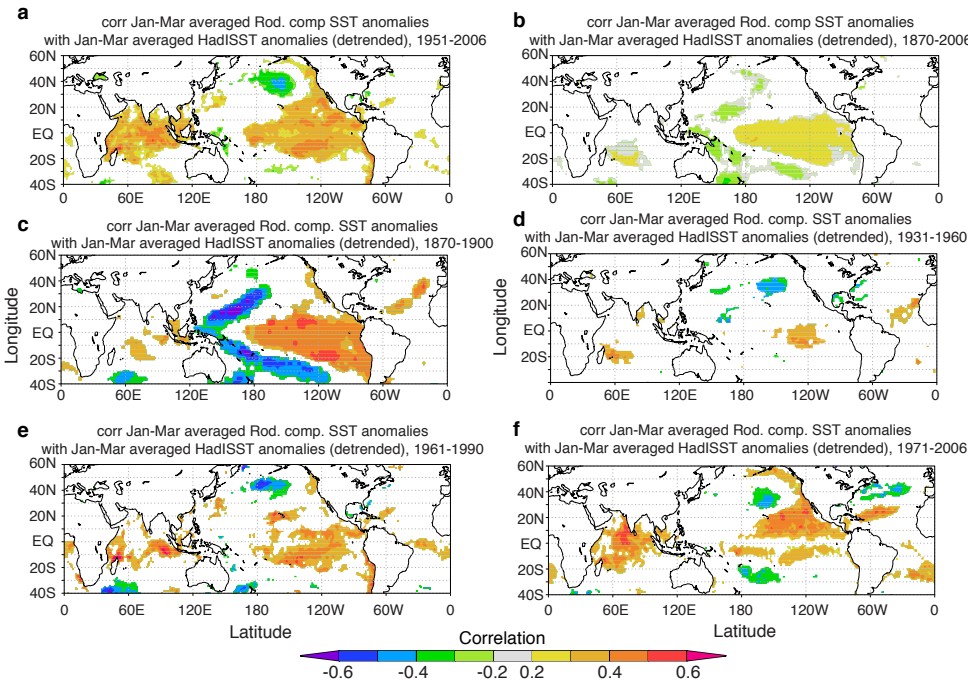


Figure A3





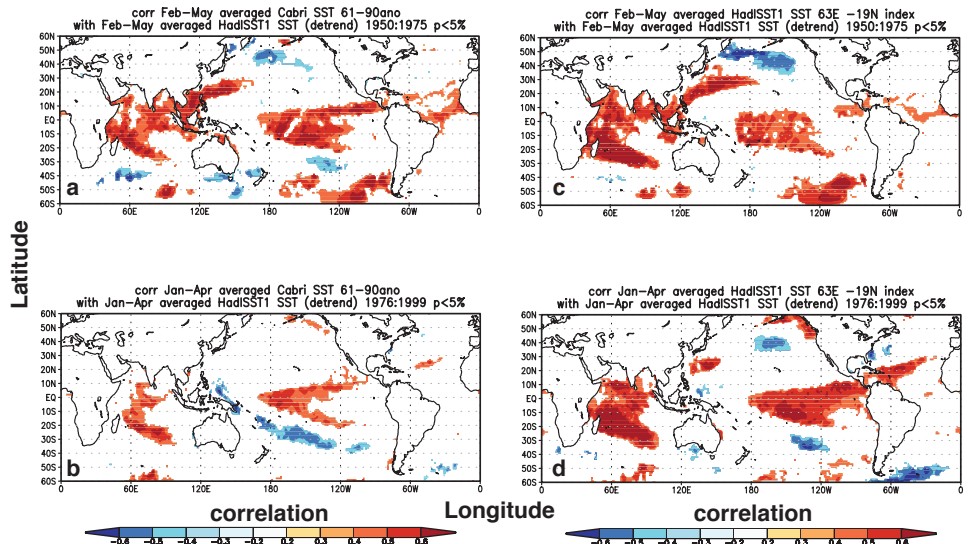


Figure A4



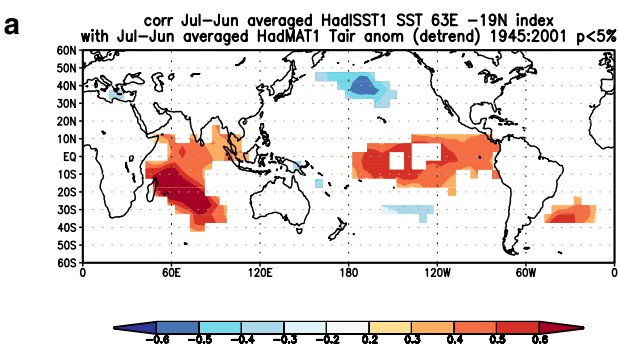

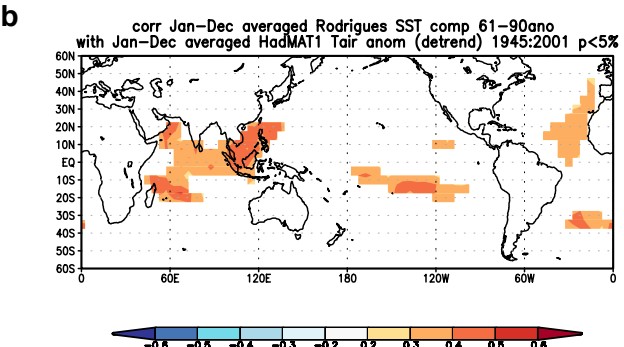

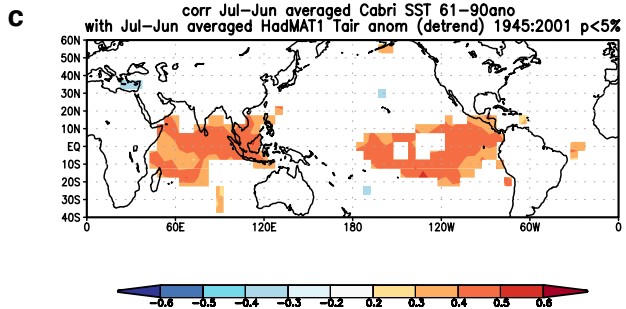


Figure A5



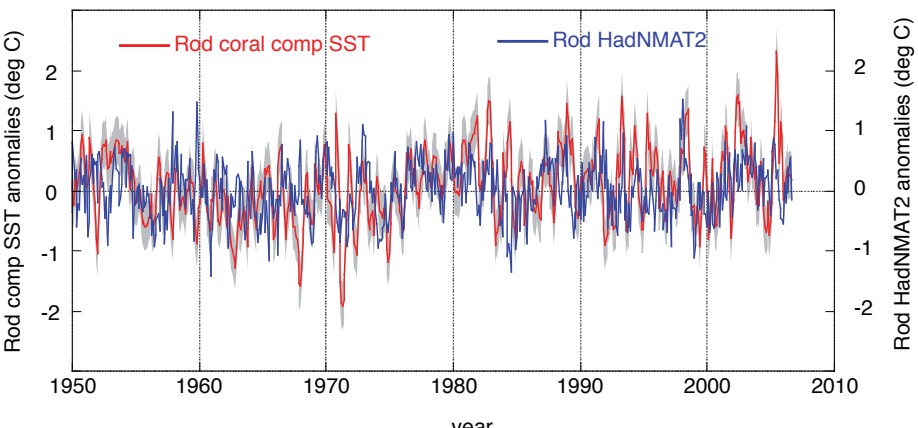


Figure A6

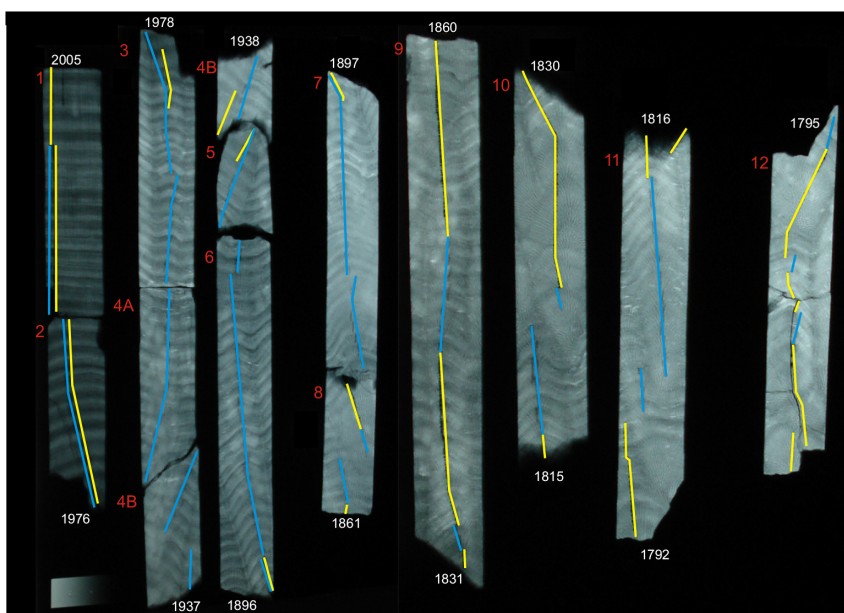


Figure A7



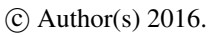

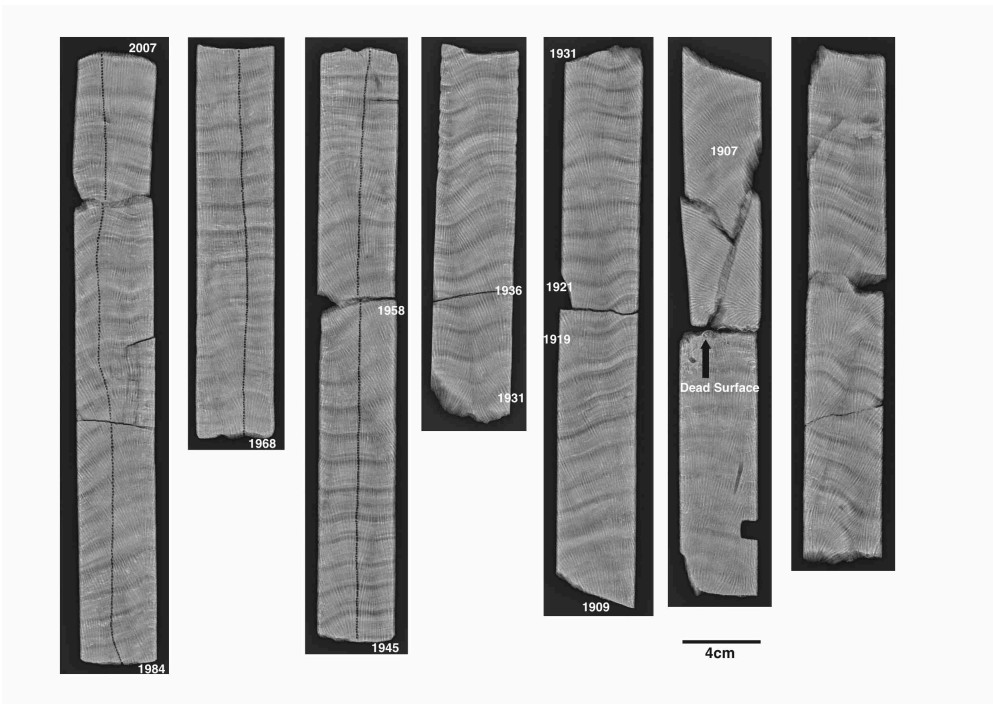


Figure A8












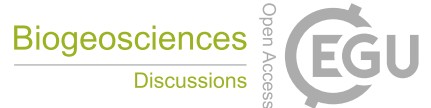

|  | SST *in situ* 2002-2006 | AVHRR SST 2002-2006 | ERSST 2002-2006 | Air Temp. 2002-2006 |
|---|---|---|---|---|
| Mean annual | 25.49 (0.24) | 25.4 (0.11) | 25.57 (0.3) | 27.49 (0.31) |
| Maximum | 28.6 (0.5) | 28.65 (0.44) | 28.29 (0.4) | 31.2 (0.62) |
| Minimum | 22.4 (0.27) | 22.75 (0.21) | 23.15 (0.13) | 24.2 (0.44) |
| Seasonal Range | 6.22 (0.68) | 5.9 (0.58) | 5.14 (0.39) | 7.0 (0.79) |
| STDV | 2.14 | 1.78 | 1.69 | 2.07 |


Table A1 – Statistics of various sea surface temperature (SST) products and air
temperature for Rodrigues with 1σ standard deviations in brackets for the period 2002 to
2006 (period with *in situ* SST data). STDV = 1σ standard deviation over all years. All
units in °C.















| (a) Max-Min | Regression equation | $r^2$ | p |
|---|---|---|---|
| Totor | Sr/Ca = -0.0439(±0.004)*SST + 10.032(±0.10) | 0.97 | <0.001 |
| Cabri | Sr/Ca = -0.0384(±0.005)*SST + 9.861(±0.12) | 0.89 | <0.001 |
| **(b) Max-Min** | | | |
| Totor | Sr/Ca = -0.0638(±0.004)*SST + 10.566(±0.09) | 0.95 | <0.001 |
| Cabri | Sr/Ca = -0.0507(±0.004)*SST + 10.179(±0.10) | 0.90 | <0.001 |
| **(c) Max-Min** | | | |
| Totor | Sr/Ca = -0.0531(±0.004)*SST + 10.271(±0.11) | 0.96 | <0.001 |
| Cabri | Sr/Ca = -0.0441(±0.005)*SST + 10.012(±0.13) | 0.88 | <0.001 |
| **(d) Monthly** | | | |
| Totor | Sr/Ca = -0.0522(±0.003)*SST + 10.272(±0.08) | 0.79 | <0.001 |
| Cabri | Sr/Ca = -0.0419(±0.003)*SST + 9.95(±0.07) | 0.87 | <0.001 |


Table A2 - Linear regression of coral Sr/Ca with a) *in situ* SST 2002-2005/6, b)
ERSSTv.3 1997-2005/6, c) AVHRR SST NOAA Coral Reef watch data 2000-2005/6 and
d) monthly Sr/Ca with AVHRR SST for the period 1982 to 2005.