# Peer review of "A sea surface temperature reconstruction for the southern Indian Ocean trade wind belt from corals in Rodrigues Island (19°S, 63°E)"

_Biogeosciences, 2016_

## Referee Comment (RC1) · Anonymous Referee #1 · 6 Apr 2016

General comments:

The authors have made new high-resolution measurements of the Sr/Ca ratios in two coral cores from Rodrigues Island in southwestern Indian Ocean. They have undertaken screening for diagenesis and detailed mapping of the corallite orientation which they apply to assess the fidelity of the coral-derived sea surface temperature (SST) reconstructions through the length of the two coral time series. They also 'calibrate' the coral Sr/Ca series with a range of available 'observational' SST and air temperature products for the region. I appreciate that available SST/air temperature products for calibration with Sr/Ca may produce different results (Section 5.3) but it is extremely confusing for the reader to distinguish what is important in the detailed dissection of the

different calibrations (e.g. Figure 6). It might be more understandable to the reader to present a summary table of the different calibrations and characteristics of the resulting SST reconstruction time series. Can we clearly identify the best 'calibration' data set for this region? At present there is a plethora of detailed descriptions but no overall synthesis or tests of whether the differences using different calibration data sets are statistically significant. Overall, I found this paper extremely hard to follow. It would greatly help if the authors clearly articulated the questions they address and then follow this framework to present the Results, Discussion and Conclusions. There is also a lack to statistical analyses whereby the fidelity/reliability of the two coral records and associated reconstructions can be objectively presented. As a consequence it is hard to determine what the main conclusions are and how well supported they are by the data and analyses presented. A shortened and more straightforward presentation of the findings could be a useful addition to the literature. I strongly recommend that the authors reconsider how they present their findings and also focus on summarising findings rather than give a detailed account of every wiggle in the time series that appears either anomalous and/or does not match the other coral or temperature data sets.

Specific comments:

Line 27: 'over recent decades' rather than 'past decades'.

Lines 29-30: 'sea surface temperature'.

Line 30: 'tropical coral reef ecosystems'.

Line 38: replace 'of paramount importance' with 'essential'.

Line 50: give the time period over which this increase was observed rather than 'the recent 15 years'.

Line 52: Do you mean the past century or the past 60 years?

Line 54: 'major role in the decrease'.

Line 59:'event causing widespread coral bleaching. . .'. Also provide reference for this statement.

Line 61: 'sustainability of tropical coral reef ecosystems'.

Line 71: 'for centuries at a rate of 0.5 and 2 cm.yr-1'.

Lines71-72: 'down-core geochemical sampling of massive corals can yield reconstructed SST time series at approximately monthly resolution.'

Line 74: 'relative to Ca, in proportion to ambient SSTs'.

Line 75: 'have been shown'.

Line 83: what is meant by 'need to be excluded by specific analysis'?

Lines 96-97: 'past variation in salinity associated with'.

Line 99: 'sea-level pressure (SLP)'.

Line 103: 'significant' rather than 'strong'.

Line 104: Does the Indian air temperature record go back to 1847?

Line 106: add (ENSO).

Line 112: Replace 'El Nino-Southern Oscillation' with 'ENSO'.

Lines 121-122: 'continuous fringing reef approximately 90 km in length'.

Lines 133-135: Confusing.

Line 136: 'in the annual mean'.

Line 141: what category of tropical cyclone is 'extreme'?

Line 142: is this referring to waves or a storm surge?

Lines 142-143: is this after they have crossed land and dissipated?

Line 144: spell out 'CTD'.

Lines 144-145: what was the sampling resolution of the CTD measurements?

Line 148: Provide the name and WMO number for the meteorological station.

Line 156: 'coral cores'.

Line 170: 'Annual density bands' rather than 'growth laminae'.

Line 172: Reorder Figures in appendix as this refers to Figs 7 and 8

Line 181: 'alteration in the Totor and Cabri cores'.

Lines 181-185: So were several slabs taken from each core? How many? Is it likely that there would be diagenesis in one slab and not another from the same core?

Line 188: What is 'RWTH'? spell out.

Line 204: 'analogous'.

Lines 212-214: Unclear how the assignment of the Sr/Ca maxima relates to the SST data.

Line 230: 'SST from sparse data'.

Line 237: 'We also used the United Kingdom Meteorological Office's monthly SST….'. Presumably the sea ice data was not relevant to this study.

Line 246: Is it relevant that this data is used by NOAA's coral reef watch program?

Lines 224-259: Suggest shortening this section and focus on the SST series actually used in the analysis. Also if average statistics of the different data sets are provided in

Appendix Table 1, there is no need to repeat in the text, just provide some general commentary about the differences/or not between the different SST products.

Line 264: values here given in mm.yr whereas in Table 1 in cm.yr.

Lines 270-285: Shorten and simplify. Is there a reason to expect long-term trends in the different growth variables.

Line 278: 'the fit is less optimal' – the fit between what and what?

Lines 290-292: is the difference in seasonality between the 2 cores significant?

Line 293: 'between average Sr/Ca ratios in the two cores. . . . . . . . .'.

Lines 297-299: Before combining the two records to form a composite series, it would be useful to know whether the two series are correlated. Also, do the SST reconstructions presented here show similar temporal variations to other coral-based climate reconstructions for the western Indian Ocean – do these new reconstructions tell us anything new or just confirm previous findings (which is a useful statement in its own right).

Lines 300-322: In the previous paragraph it was indicated that the Sr/Ca ratios were converted to SST – why not present the SST reconstructions in Figure 3 and use these in the text rather than having to explain that more negative = warming etc? Also suggest simplifying this section as it is hard to determine what the authors are trying to convey apart from identifying wiggles in the time series. How about including some statistics, e.g. are there significant linear trends etc? Also suggest including a weighted filter in the time series graphs to illustrate the decadal variability referred to.

Lines 324-394: I found this section very long and confusing. Why not present the SST:calibrations first in the Results section and then go on to discuss what the SST reconstructions tell us about SST variability in the region? It would be worth considering moving some of the details of the calibration methods to Supplementary Material – as a reader I am getting lost as to what was done and why.

Line 388: What is meant by 'slab 7'? Is this s different slice from the coral or is it the section number downcore?

Line 395: I have stopped commenting at this point on writing style and clarification.

Lines 396-424: Should this section on diagentic alterations not come earlier in the Results section?

Lines 426-453: Again, this is long-winded and confusing for the reader. What questions are being addressed by the authors in this section? How as 'Indian Ocean wide SST' calculated and with what data set of the many used in this study?

Lines 456-536: Again this section is long-winded and confusing for the reader. It is very hard to determine what the main discussion points are.

Lines 538-617: Comments as for the previous sections – confused and confusing and hard to determine what is being done and why.

Lines 619-635: The main conclusion seems to me to be the need for careful screening of coral samples (for diagenesis and corallite orientation) before measuring and developing climate reconstructions. Do the reconstructions actually tell us anything new about SST variability in the Indian Ocean given the main period is 1945-2006?

Line 864 (Table 1): clarify that depth is of the colony; also provide time periods for the calculations of average growth characteristics.

Line 882: 'Rodrigues Island'

Lines 887-889: provide the sampling resolution for these graphs.

Lines 892-894: Indicate in Figure caption that y-scale for Sr/Ca is inverted.

Lines 936-939: There are 3 panels to Fig A1 – explain each in caption; also indicate temporal resolution of time series.

Supplementary Tables 1-26: This is a lot of information that I feel needs to be better synthesised for the reader. Also, in carrying out so many correlations on the same set of time series, has any allowance been made for loss of degrees of freedom? i.e. the number of correlations undertaken increases the probability of obtaining a significant correlation by chance.

[Figure]

---

## Referee Comment (RC2) · Anonymous Referee #2 · 3 May 2016

General comments:

This paper presents two new Sr/Ca-SST reconstructions from Rodrigues Island in the south-central Indian Ocean, which contribute to our understanding of SST variability and trends in this region and their relationship to modes of climate variability (PDO, ENSO). The authors present a very rigorous assessment of the potential impact of diagenesis, corallite orientation, and Sr/Ca-SST calibration on these Sr/Ca-SST reconstructions. The results of this analysis emphasize the importance of corallite orientation and screening for diagenesis in such reconstructions, as suggested in previous work by DeLong et al. (2012), Hendy et al. 2007 (and others). However, a number of warm and cold anomalies may not be completely explained by either corallite angle or diagenesis, and the authors conclude that the SST reconstruction is only reliable back to 1945. This leaves little gained beyond the instrumental record, though additional support for the importance of these issues is still an important contribution on its own. Nonetheless, I have a number of other major concerns that need to be addressed by the authors before publication:

1) Calculation of the composite: the composite was calculated by taking the arithmetic mean of the coral records from each site, yet the authors do not demonstrate strong agreement between the two coral records before compositing. The authors need to show statistics supporting the agreement between the records. E.g., what is the correlation between the two records? Based on figure 3, there appears to be disagreement between the two records such that when averaged, the variability of the composite is reduced over the interval that included the two records (relative to that of the earlier period when only 1 record is available). The two records also have opposing trends over the 1951-2005 interval (as discussed on lines 359-363)! The moderate trend of the composite (0.44 degrees) is simply a result of averaging the strong positive (1.38 degrees) and moderate negative trend (-0.49 degrees) and thus isn't physically interpretable. The climate signal also appears to be weakened in the composite (e.g., Figure A5).

2) Selection of Sr/Ca-SST calibration: the authors compute a local calibration with both in situ and gridded SST data, but then use the relationship from Corrège 2006. The justification for this is not clear from the paper. Since local SST data is available, the authors should use this calibration unless they have a valid reason not to use the in situ data.

3) Sr/Ca-SST calibration methodology: the authors should use a reduced major axis regression instead of simple linear regression to calibrate their Sr/Ca records with SST. RMA takes into account errors in both SST and Sr/Ca, which is critical given that the SST observations themselves are also imperfect (see Solow and Huppert 2004; York et al. 2004; Thirumalai et al. 2011). It is also unclear why the authors use only the

max and min to calibrate, rather than the full record of monthly anomalies over the calibration period. Justification for this choice is needed.

4) Potential warm and cool biases in the coral records: It is unclear which of the warm and cool biases (highlighted in figure 6 and discussed on lines 414-424) were included in the composite, and which were removed. From the composite of figure 6, it looks like many of them were averaged in. Clearer justification of their inclusion is needed. Given the magnitude of these anomalies and the fact that their source is unclear (in the cases where no clear diagenesis was identified), the authors should investigate whether removing these events from the record changes their results.

Specific comments:

Lines 289-292: it is difficult to see this comparison of seasonality from figure 3. I suggest showing the period of overlap separately to demonstrate the agreement between the records

Lines 301-303: Figure 3 does not effectively portray the trends discussed here

Lines 325-327: Discuss these calibration methods earlier (when discussing the calibration approach in the methods section)

Line 329: This validation period includes part of the calibration period. Stop in 2002 to have independent calibration/validation periods.

Lines 374-394: This comparison with SST over the past 150 years is not a very useful exercise given the paucity of data at this site (as shown in figure A2). The authors seem to be using the agreement with the instrumental data over the full record to support their reconstruction, but this reasoning is circular (we need a coral reconstruction because there aren't enough observations, but then we use the observations to validate our record). Stick to the well observed period for the calibration/validation exercise. It is very possible that some of these discrepancies between the Sr/Ca-SST and SST are due to biases in the SST record.

Line 412: see also Sayani et al. 2011

Lines 412-413: What about dissolution—any indication that dissolution could explain these discrepancies (e.g., see Sayani et al. 2011—who show that dissolution is associated with low Sr/Ca anomalies/warm biases)?

Lines 434-436: Recommend performing a running correlation analysis to test this.

Lines 522-526: other potential drivers of these discrepancies? E.g., see Alpert et al. 2015

Line 602-607: corals may have acclimatized or adapted to the high temperature variability. A number of studies have shown that corals in sites that have high temperature variability may be less susceptible to bleaching (e.g., Thompson and van Woesik 2009, Donner 2011). This variability is now usually taken into account when calculating the thermal stress thresholds to predict bleaching (e.g., Kleypas et al. 2015), as this approach has been shown to better predict observed bleaching patterns (e.g., Logan et al. 2012); this should be used instead of the conventional degree heating weeks threshold on lines 605-607.

Lines 611-612: citation?

Figure 8 caption: this caption needs to be reworded. It is hard to follow what is in each panel.

Figure 9: why divide the analysis into these periods? This needs to be justified somewhere. If the goal was to compare among different phases of the PDO & ENSO (which from the text appears to be the goal), then the authors should select periods that line up with the phases of these modes.

Figure 2: change the color scheme so that the lines in 2a are differentiable

Figure 7: the markings denoting corallite angle are not clear—where do the transitions occur, at the end of the lines? May be clearer if brackets are used to denote the

sections with different corallite angles

Figure A1: change the color scheme so that the black lines are differentiable

Figure A4: what is the difference between the figures on the left and right? This is not indicated in the caption, and it is not clear from the figure. Clarify in caption.

Technical corrections:

Lines 384-387: reword

Line 432: long-term

Line 548: closest agreement to

Table A1 caption, line 977: change "in brackets" to "in parentheses"

References:

Alpert, Alice E., et al. Comparison of equatorial Pacific sea surface temperature variability and trends with Sr/Ca records from multiple corals. Paleoceanography (2016).

Donner, Simon D. An evaluation of the effect of recent temperature variability on the prediction of coral bleaching events. Ecological Applications 21.5 (2011): 1718-1730.

Logan, Cheryl A., et al. Incorporating adaptive responses into future projections of coral bleaching. Global Change Biology 20.1 (2014): 125-139.

Kleypas, Joan A., et al. The impact of ENSO on coral heat stress in the western equatorial Pacific. Global change biology 21.7 (2015): 2525-2539.

Sayani, Hussein R., et al. Effects of diagenesis on paleoclimate reconstructions from modern and young fossil corals. Geochimica et Cosmochimica Acta 75.21 (2011): 6361-6373.

Solow, A. R., & Huppert, A. A potential bias in coral reconstruction of sea surface temperature. Geophysical research letters 31.6 (2004).

Thirumalai, K., A. Singh, and R. Ramesh. A MATLAB[TM] Code to Perform Weighted Linear Regression with (correlated or uncorrelated) Errors in Bivariate Data, Journal Of The Geological Society of India, 77(April), (2011): 377-380.

Thompson, D. M., and R. Van Woesik. Corals escape bleaching in regions that recently and historically experienced frequent thermal stress. Proceedings of the Royal Society of London B: Biological Sciences 276.1669 (2009): 2893-2901.

York, D., N.M. Evensen, M.L. Martínez., and J.D.B. Delgado. Unified equations for the slope, intercept, and standard errors of the best straight line, American Journal of Physics, 72 (2004): 367.
* * *

---

## Author Comment (AC1) · 27 May 2016

Response to Reviewer comments on Biogeosciences Discuss., doi:10.5194/bg-2016-69, 2016 by Zinke et al.

We would like to thank both reviewers for their assessment of our research article. We are particularly grateful for the constructive suggestions provided. Below, we address the major concerns and suggestions of both reviewers. We hope that our response merits revision of our manuscript for publication in Biogeosciences.

1) Response to Reviewer 1 in terms of clarity of manuscript

We agree with Reviewer 1 that we could more clearly articulate the main questions to

be addressed at the end of the Introduction and present the Results, Discussion and Conclusions following this framework.

Abstract of revised version:

"Here, we aim to reconstruct past SSTs from Sr/Ca ratios in two coral cores obtained from Rodrigues Island (19°S, 63°E) located 690 km to the East of Mauritius within the trade wind belt of the south-central Indian Ocean. We assess relationships between the observed long-term SST and climate fluctuations related to the El Nino-Southern Oscillation (ENSO) and the Pacific Decadal Oscillation (PDO) between 1945 and 2006, respectively. To obtain a robust SST record, we assess the reproducibility of the Sr/Ca proxy from two different locations, and we provide a rigorous assessment of the potential impacts of diagenesis and corallite orientation on Sr/Ca-SST reconstructions. We calibrate individual robust Sr/Ca records with in-situ SST and various gridded SST products. The results show that our reliable SST record from Carbi provides the first Indian Ocean coral proxy time series that records the SST signature of the PDO in the SW subtropical Indian Ocean since 1945."

Response to Reviewer 1 and 2: Use of coral composites

We agree with both Reviewers that the calculation of the coral composite needs to be verified by showing agreement between the individual records. We have therefore decided to omit the calculation of a composite record since the climate signals also appear to be weakened as compared to our record from Cabri (based on agreement on annual, interannual time scales with SST, ENSO and PDO). Instead, we will focus the climatic interpretation on the Cabri record that extends from 1945 to 2006 and will adjust the Figures accordingly.

Response to Reviewer 1 and 2: Statistical analysis

We will omit all correlations using the coral composite since we will no longer attempt to compute a composite record. We have performed a number of correlations illustrated

in Table S1 to S26. All significance levels of the correlations take into account the loss of degrees of freedom. A 95% confidence interval based on a Monte Carlo simulation is also indicated in the Tables that shows the upper and lower bound of correlation coefficients (see knmi website and reference Trouet, V. and van Oldenborgh, G.J. KNMI Climate Explorer: a web-based research tool for high-resolution paleoclimatology. Tree Ring Research 69, 1, 3-13 (2013). We will perform the running correlation analysis as requested by reviewer 2 to support lines 434-436 and show them as Supplementary Figure.

Corallite orientation

Reviewer 1 found our rigorous assessment of effects of corallite orientation on SSTs rather long-winded and confusing. This is in contrast to Reviewer 2 who emphasized the importance of such a 'rigorous assessment' to provide reliable SST reconstructions. We agree with DeLong et al. (2012), who were the first to make such an assessment, that examining effects of corallite orientation proves useful and makes the SST series more robust. We focused on our long SST record from Totor, pre-1945. It illustrates the importance of having multiple Sr/Ca records from nearby regions as corallite orientation from one record may bias the SST reconstruction. We will archive the data for both cores on the NOAA WDC paleodata webarchive which will enable comparison of the Totor Sr/Ca record with future records. Our rigorous assessment of potential reconstruction biases will therefore provide the user with important information which otherwise would be lost to the proxy community when performing larger scale analysis based on multiple records.

Use of multiple instrumental data products

As mentioned by reviewer 2, SST observations from gridded products are extremely sparse for our region and one could argue that discrepancies between proxy and SST data simply arise from a lack of observations. At present, it is not clear which gridded SST data are best suited for the Indian Ocean and tropical oceans in general. The

use of multiple SST products is now a standard procedure in almost all meteorological studies. We have adopted this approach in our manuscript. In our opinion it is therefore extremely important to assess/illustrate the agreement between various SST products for our region with our proxy data. We do, however, agree with Reviewer 1 that we should summarize which SST dataset appears to agree best with our reconstruction.

Response to Reviewer 1 and 2 comments on information gained by new proxy record

Reviewer 1 asked if the reconstructions tell us 'anything new about SST variability in the Indian Ocean" and reviewer 2 asked what climate information we gain from the proxy record beyond the instrumental record. The main results (as stated in abstract, discussion and conclusions) from our study is that the Cabri Sr/Ca record provides the first SST reconstruction from the tropical and subtropical Indian Ocean that shows a clear relationship between SST fluctuations and the PDO since 1945. Previous studies have shown only indirect links between the PDO with sea level pressure and salinity (Crueger et al., 2009), hydrological balance (Zinke et al., 2008) and river runoff (Grove et al., 2013) in the western Indian Ocean. In addition, our record includes a Sr/Ca record, which is currently the most reliable proxy for SST in corals. The only long record from this region of the Indian Ocean is a stable isotope record from Reunion Island that mainly records salinity variations. Therefore, our new proxy record from Rodrigues for the period between 1945 and 2006 is a valuable addition to the sparse Indian Ocean coral proxy network. It also provides us with the knowledge that records from Rodrigues are well suited when studying climate teleconnections with the PDO (as stated in abstract and conclusions). Even the long record might be proven invaluable as a subtropical Indian Ocean record in the near future. We demonstrate that the Totor record does follow grid-SST in the 19th and early 20th century for several decades. Only further replication with Sr/Ca records from the same site or nearby sites can provide further validation of the long Totor Sr/Ca record. Crueger, T., Zinke, J. and Pfeiffer, M. 2009. Dominant Pacific SLP and SST variability recorded in Indian Ocean corals. International Journal of Earth Sciences 98, Special Volume. doi:10.007/s00531-008-

0324-1. Grove, C. A., Zinke, J., Peeters, F., Park, W., Scheufen, T., Kasper, S., Randria-manantsoa, B., McCulloch, M. T. and Brummer, GJA 2012. Madagascar corals reveal Pacific multidecadal modulation of rainfall since 1708. Climate of the Past 9, 641-656. Zinke, J., Timm, O., Pfeiffer, M., Dullo, W.-Chr., Kroon, D. and Thomassin, B. A. 2008. Mayotte coral reveales hydrological changes in the western Indian between 1865 to 1994. Geophysical Research Letters 35, L23707, doi:10.1029/2008GL035634.

Response to Reviewer 2 point 2) on Selection of Sr/Ca-SST calibration

We compute calibrations with local and regional grid-SST data over a short time interval 2002 to 2006 and with satellite SST/grid-SST back to 1981 (Table A2). The local calibration with in-situ SST is based on four years only and it is currently not known if the slope of the short calibration period would be stable over longer periods. The application of the regression slope for the entire record is therefore not robust. Corrège (2006) provided regression slopes from a greater density of calibrated records across the global tropics and his 'global' slope of $\sim$ -0.06mmol/$^\circ$C agrees with the range of slopes that we obtained (Table A2). Most coral records used in Corrège (2006) were calibrated with satellite or grid-SST. Since we are interested in the reconstruction of large-scale southern Indian Ocean SST and their teleconnections with global climate modes, we use of grid-SST. We account for the full spread in regression slopes reported in the literature (-0.4 to -0.084 mmol/$^\circ$C) and our uncertainty bounds are rather conservative estimates. In addition, it has been shown by Nurhati et al. (2011) that reconstructions of absolute SST have large errors (up to 7$^\circ$C) while those of relative SST (anomalies) are lower (<1$^\circ$C). Therefore, in our study we use SST anomalies calculated with the mean slope from Corrège (2006) for the assessment of interannual climate relationships. Corrège, T., Sea surface temperature and salinity reconstruction from coral geochemical tracers. Palaeogeo. Palaeoclim. Palaeoeco., 232, 408-428, 2006. Nurhati, I. S., K. M. Cobb and E. D. Lorenzo (2011). Decadal-Scale SST and Salinity Variations in the Central Tropical Pacific: Signatures of Natural and Anthropogenic Climate Change. Journal of Climate 24: 3294-3308.

Response to Reviewer 2 point 3) on Sr/Ca-SST calibration method

We decided to use ordinary least squares (OLS) regression for the calibration of our coral records, as this is the method best suited for asymmetric relationships. The coral Sr/Ca-SST relationship is clearly asymmetric (SST influences coral Sr/Ca, coral Sr/Ca has no influence on SST). A potential error in the instrumental data does not justify the use of RMA. See Smith (2009): Use and misuse of the reduced major axis for line-fitting (DOI: 10.1002/ajpa.21090) for a discussion. Solow and Huppert (2004) (incorrectly cited by the reviewer as suggesting RMA regression for coral calibrations) also advocate the use of OLS for the calibration of coral proxies. They do not recommend RMA regression: The biggest problems with the application of RMA for coral-Sr/Ca calibrations are the unknown errors. RMA assumes that the error variance in the SST observations equals the error variance of the Sr/Ca determinations. There is no reason to believe that this assumption is warranted. The RMA method can be extended to allow for differences in the error variances. To do so, it is necessary to have an estimate of both the SST and Sr/Ca error variance. However, it is practically impossible to determine the error variance of coral Sr/Ca determinations, as these include not only the analytical error but also other factors such as vital effects or skeletal heterogeneities. Nevertheless, we do not reconstruct absolute SST for the entire time series. Instead we reconstruct relative SST changes or SST anomalies, which have a much lower error than absolute SST estimates (see Nurhati et al., 2011). The calibration exercise is provided in order to give the reader an idea how well absolute SST is recorded for the 4 years of in-situ SST measurements. We report the various calibrations since this is now standard procedure.

Response to Reviewer 1 and 2: questions about diagenesis section

Reviewer 1: Line 83/181-185/388: We now consistently use core section instead of slab. Line 412-413: Reviewer 2 asks: "Any indication that dissolution could explain these discrepancies?" Thin sections and SEM studies are often used to detected dissolution in reef corals (Hendy et al., 2007; McGregor and Abram, 2008; Sayani et al.,

2011). The application of both techniques in this study showed that the two coral cores are devoid of dissolution. Hendy et al (2007) showed that dissolution during marine diagenesis leads to an increase in Sr/Ca and therefore an apparently cold temperature anomaly. Dissolution during marine diagenesis therefore would not be able to cause the observed positive temperature anomaly. Decreased Sr/Ca values in diagenetically modified corals have been attributed to aragonite dissolution and concomitant calcite cementation in a meteoric environment (Sayani et al., 2011). With a combination of SEM, thin section microscopy and XRD we demonstrated the lack of dissolution and calcite cementation in the corals and therefore can rule out any influence of dissolution on the proxy record. Hendy, E. J., Gagan, M. K., Lough, J. M., McCulloch, M., and deMenocal P. B.: Impact of skeletal dissolution and secondary aragonite on trace element and isotopic climate proxies in Porites corals, Paleoceanography, 22, PA4101, doi:10.1029/2007PA001462, 2007. McGregor, H. V. and Abram, N. J.: Images of diagenetic textures in Porites corals from Papua New Guinea and Indonesia, Geochemistry, Geophysics, Geosystems 9(10), doi:10.1029/2008GC002093, 2008. Sayani, H. R., Cobb, K. M., Cohen, A. L., Crawford Elliott, W., Nurhati, I. S., Dunbar, R. B., Rose, K. A., Zaunbrecher, L. K.: Effects of diagenesis on paleoclimate reconstructions from modern and young fossil corals, Geochimica et Cosmochimica Acta, 75, 6361–6373, 2011.

Modified caption to Fig. 8: Thin-section and SEM images of pristine coral skeleton and diagenetic alteration in cores Totor and Capri. A and B: Excellent preservation of the primary coral aragonite (PA) in core Totor. Trace amounts of aragonite cements (AC) occur as isolated patches in core sections 6 (C), 7 (D) and 11 (E) of Totor. F (left): A prominent growth break (stippled line) in core section 12 of Totor is encrusted by coralline red algae (CRA). F (middle): The section above the growth break is well preserved. F (right): The pristine coral skeleton of core Capri locally contains aragonitic sediment (S) partially filling pore spaces. Thin section photographs are shown in plane- (left) and cross-polarized light (middle).

---

## Author Response (AR1)

**Institut für Geowissenschaften,**
**Sektion Paläontologie,**
**Malteserstrasse 74-100,**
**12249 Berlin, Germany**

Freie Universität Berlin, Institut für Geowissenschaften,
Section Paleontology
Malteserstrasse 74-100, 12249 Berlin

To
Editor Biogeosciences
Natascha Töpfer
Copernicus Publications

Dr. Jens Zinke
Sektion Paläntologie
12249 Berlin

| | |
|---|---|
| **Telefon** | +49 30 838-61034 |
| **Fax** | +49 30 838-70745 |
| **E-Mail** | jzens.inke@fu-berlin.de |
| **Internet** | www.fu-berlin.de |

16.08.2016

**Revision bg-2016-69**

Dear Editor,

Please find enclosed our revised manuscript bg-2016-69 entitled "A sea surface temperature reconstruction for the southern Indian Ocean trade wind belt from corals in Rodrigues Island (19°S, 63°E)", for consideration as a research article in Biogeosciences.

We thank both reviewers for their constructive comments that helped us to improve the mansucript. You will find our detailed response in the document 'Point-by-point response to reviewers comments' that we have added to this letter.

We sincerely hope that our revised mansucript is now suitable for publication in Biogeosciences.

Kind regards,
Jens Zinke

**Point by Point response to Reviewers' comments:**

**Anonymous Referee #1**

General comments:

The authors have made new high-resolution measurements of the Sr/Ca ratios in two coral cores from Rodrigues Island in southwestern Indian Ocean. They have undertaken screening for diagenesis and detailed mapping of the corallite orientation which they apply to assess the fidelity of the coral-derived sea surface temperature (SST) reconstructions through the length of the two coral time series. They also 'calibrate' the coral Sr/Ca series with a range of available 'observational' SST and air temperature products for the region. I appreciate that available SST/air temperature products for calibration with Sr/Ca may produce different results (Section 5.3) but it is extremely confusing for the reader to distinguish what is important in the detailed dissection of the different calibrations (e.g. Figure 6). It might be more understandable to the reader to present a summary table of the different calibrations and characteristics of the resulting SST reconstruction time series. Can we clearly identify the best 'calibration' data set for this region? At present there is a plethora of detailed descriptions but no overall synthesis or tests of whether the differences using different calibration data sets are statistically significant. Overall, I found this paper extremely hard to follow. It would greatly help if the authors clearly articulated the questions they address and then follow this framework to present the Results, Discussion and Conclusions. There is also a lack to statistical analyses whereby the fidelity/reliability of the two coral records and associated reconstructions can be objectively presented. As a consequence it is hard to determine what the main conclusions are and how well supported they are by the data and analyses presented. A shortened and more straightforward presentation of the findings could be a useful addition to the literature. I strongly recommend that the authors reconsider how they present their findings and also focus on summarising findings rather than give a detailed account of every wiggle in the time series that appears either anomalous and/or does not match the other coral or temperature data sets.

**We agree with Reviewer 1 that we could more clearly articulate the main questions to be addressed at the end of the Introduction and present the Results, Discussion and Conclusions following this framework.**

**The end of the Introduction in the revised version now reads:**

"Here, we aim to reconstruct past SSTs from Sr/Ca ratios in two coral cores obtained from Rodrigues Island (19°S, 63°E) located 690 km to the North-East of Mauritius within the trade wind belt of the south-central Indian Ocean. To obtain a robust SST record, we assess the reproducibility of the Sr/Ca proxy, and we provide a rigorous assessment of the potential impacts of diagenesis and corallite orientation on Sr/Ca-SST reconstructions. We calibrate individual Sr/Ca records with *in-situ* SST and various gridded SST products and verify the suitability of SST products for climate studies in the south-central Indian Ocean. Furthermore, we assess relationships between the observed long-term SST and climate fluctuations related to the El Nino-Southern Oscillation (ENSO), the Suptropical Indian Ocean Dipole Mode (SIOD) and the Pacific Decadal Oscillation (PDO) between 1945 and 2006, respectively."

In addition, the paper has been shortened considerably. Our findings regarding anomalous coral Sr/Ca values and corallite orientation have been summarized in a table, and a large part of the discussion has been moved to the supplementary material. Only our key findings are discussed in the main manuscript (see section 6.1).

Specific comments:

Line 27: 'over recent decades' rather than 'past decades'.

**Done.**

Lines 29-30: 'sea surface temperature'.

**Done.**

Line 30: 'tropical coral reef ecosystems'.

**Done.**

Line 38: replace 'of paramount importance' with 'essential'.

**Done.**

Line 50: give the time period over which this increase was observed rather than 'the recent 15 years'.

**Done.**

Line 52: Do you mean the past century or the past 60 years?

**The past 60 years as stated**

Line 54: 'major role in the decrease'.

**Done.**

Line 59:'event causing widespread coral bleaching. . .'. Also provide reference for this statement.

**Done, Sheppard, 2003 added.**

**Done.**

Line 61: 'sustainability of tropical coral reef ecosystems'.

**Done.**

Line 71: 'for centuries at a rate of 0.5 and 2 cm.yr-1'.

**Done.**

Lines71-72: 'down-core geochemical sampling of massive corals can yield reconstructed SST time series at approximately monthly resolution.'

**Done.**

Line 74: 'relative to Ca, in proportion to ambient SSTs'.

**Done.**
Line 75: 'have been shown'.

**Done.**
Line 83: what is meant by 'need to be excluded by specific analysis'?

**Changed to 'specific petrographic analysis'**

Lines 96-97: 'past variation in salinity associated with'.

**Done.**
Line 99: 'sea-level pressure (SLP)'.

**Done.**
Line 103: 'significant' rather than 'strong'.

**Done.**
Line 104: Does the Indian air temperature record go back to 1847?

**Yes, it's the record from the India Meterological Bureau**

Line 106: add (ENSO).

**Done.**
Line 112: Replace 'El Nino-Southern Oscillation' with 'ENSO'.

**Done.**
Lines 121-122: 'continuous fringing reef approximately 90 km in length'.

**Done.**

Lines 133-135: Confusing.

**It is unclear to the authors why this is confusing.  The sentences are clear and make sense.**

Line 136: 'in the annual mean'.

**Done.**
Line 141: what category of tropical cyclone is 'extreme'?

**We added in brackets: "(category 3 and higher)", strength of extremes also explained at the end of the same sentence: "with winds of 280 km/h and waves that reach 100 m inland and 2 m above sea level. They usually last five to ten days (Turner and Klaus, 2005)."**

Line 142: is this referring to waves or a storm surge?

**Storm surge, now added**
Lines 142-143: is this after they have crossed land and dissipated?

**Yes**

Line 144: spell out 'CTD'.

**Done.**

Lines 144-145: what was the sampling resolution of the CTD measurements?

**hourly**

Line 148: Provide the name and WMO number for the meteorological station.

**WMP 61988 added, (name: Rodrigues, Mauritius)**

Line 156: 'coral cores'.

**Done.**

Line 170: 'Annual density bands' rather than 'growth laminae'.

**Done.**

Line 172: Reorder Figures in appendix as this refers to Figs 7 and 8

**Done.**

Line 181: 'alteration in the Totor and Cabri cores'.

**Done.**

Lines 181-185: So were several slabs taken from each core? How many? Is it likely that there would be diagenesis in one slab and not another from the same core?

**Only 1 slab for each core were taken**

Line 188: What is 'RWTH'? spell out.

**Rheinisch-Westfälische Technische Hochschule, now added**

Line 204: 'analogous'.

**Done.**

Lines 212-214: Unclear how the assignment of the Sr/Ca maxima relates to the SST data.

**Added 'Sr/Ca-maxima' in brackets**

Line 230: 'SST from sparse data'.

**Done.**

Line 237: 'We also used the United Kingdom Meteorological Office's monthly SST. ..'. Presumably the sea ice data was not relevant to this study.

**This is the name of the dataset and correctly stated**

Line 246: Is it relevant that this data is used by NOAA's coral reef watch program?

**Deleted.**

Lines 224-259: Suggest shortening this section and focus on the SST series actually used in the analysis. Also if average statistics of the different data sets are provided in

Appendix Table 1, there is no need to repeat in the text, just provide some general commentary about the differences/or not between the different SST products.

**All data are used to show agreement/disagreement with proxy data and between SST products. This is essential because the optimal SST product for use in climate modelling and paleoclimate studies has yet to be determined.**

**We now refer to Appendix Tab. 1 for actual statistics of SST time series and omitted repetition in text.**

Line 264: values here given in mm.yr whereas in Table 1 in cm.yr.

**Changed to mm/yr in both**

Lines 270-285: Shorten and simplify. Is there a reason to expect long-term trends in the different growth variables.

**To our opinion, it is important to provide the growth characteristics. There is no *a-priori* assumption of long-term trends between growth variables.**

Line 278: 'the fit is less optimal' – the fit between what and what?

**Changed to: "The density banding is obscured between 1877 and 1907 due to the dead surface in Cabri."**

Lines 290-292: is the difference in seasonality between the 2 cores significant? Line 293: 'between average Sr/Ca ratios in the two cores. . .. . .. . .'.

**Changed to: "…yet the difference is statistically not significant (both overlap within 1σ). "**

Lines 297-299: Before combining the two records to form a composite series, it would be useful to know whether the two series are correlated. Also, do the SST reconstructions presented here show similar temporal variations to other coral-based climate reconstructions for the western Indian Ocean – do these new reconstructions tell us anything new or just confirm previous findings (which is a useful statement in its own right).

**We no longer attempt to composite the two time series. Instead, we focus on core Cabri for evaluating the climatic signals recorded at Rodrigues (section 6.2). Core Cabri shows the SST signature of the PDO. This is a novel finding. Previous coral records from the south-western Indian Ocean recorded PDO-related variations in river runoff (Grove et al., 2013), salinity (Pfeiffer et al., 2004) and sea level pressure (Crüger et al., 2009). Figure 9 compares core Cabri and a coral Sr/Ca record from Madagascar – these two are reasonably well correlated. Please see also our response to the reviewer comments regarding Lines 619-635.**

Lines 300-322: In the previous paragraph it was indicated that the Sr/Ca ratios were converted to SST – why not present the SST reconstructions in Figure 3 and use these in the text rather than having to explain that more negative = warming etc? Also suggest simplifying this section as it is hard to determine what the authors are trying to convey apart from identifying wiggles in the time series. How about including some statistics, e.g. are there significant linear trends etc? Also suggest including a weighted filter in the time series graphs to illustrate the decadal variability referred to.

**We combined former Fig. 3 with Fig. 6 and show in the new Figure 2 a) the original Sr/Ca time series as routinely required in any such coral proxy study, b) converted SST record for Cabri and c) converted SST record for Totor.**

Lines 324-394: I found this section very long and confusing. Why not present the SST:calibrations first in the Results section and then go on to discuss what the SST reconstructions tell us about SST variability in the region? It would be worth considering moving some of the details of the calibration methods to Supplementary Material – as a reader I am getting lost as to what was done and why.

**We restructured the Results section.**

Line 388: What is meant by 'slab 7'? Is this s different slice from the coral or is it the section number downcore?

**It refers to the section of the core. We now consistently use the term 'core section'**

**instead of slab where it is appropriate.**

Line 395: I have stopped commenting at this point on writing style and clarification.

Lines 396-424: Should this section on diagentic alterations not come earlier in the Results section?

**This section has been moved into the Results section.**

Line 412-413: Any indication that dissolution could explain these discrepancies … ?

**Thin sections and SEM are often used to detect dissolution in reef corals (Hendy et al., 2007; McGregor and Abram, 2008; Sayani et al., 2011). The application of both techniques in this study showed that the two coral cores are devoid of dissolution. Hendy et al (2007) showed that dissolution during marine diagenesis leads to an increase in Sr/Ca and therefore an apparently cold temperature anomaly. Dissolution during marine diagenesis therefore would not be able to cause the observed positive temperature anomaly. Decreased Sr/Ca values in diagenetically modified corals have been attributed to aragonite dissolution and concomitant calcite cementation in a meteoric environment (Sayani et al., 2011). With a combination of SEM, thin section microscopy and XRD we demonstrated the lack of dissolution and calcite cementation in the corals and therefore can rule out any influence of dissolution on the proxy record.**

Lines 426-453: Again, this is long-winded and confusing for the reader. What questions are being addressed by the authors in this section? How as 'Indian Ocean wide SST' calculated and with what data set of the many used in this study?

**Though it is not clear what the reviewer does not understand the text was revised to make the subjects of the sentence clear.**

Lines 456-536: Again this section is long-winded and confusing for the reader.

**Though it is not clear what the reviewer does not understand the text was revised to make the subjects of the sentence clear.**

It is very hard to determine what the main discussion points are.

Lines 538-617: Comments as for the previous sections – confused and confusing and hard to determine what is being done and why.

**Though it is not clear what the reviewer does not understand the text was revised to make the subjects of the sentence clear.**

Lines 619-635: The main conclusion seems to me to be the need for careful screening of coral samples (for diagenesis and corallite orientation) before measuring and developing climate reconstructions. Do the reconstructions actually tell us anything new about SST variability in the Indian Ocean given the main period is 1945-2006?

**The main results (as stated in abstract, discussion and conclusions) from our study is that the Cabri Sr/Ca record provides the first SST reconstruction from the tropical and subtropical Indian Ocean that shows a clear relationship between SST fluctuations and the PDO since 1945. Previous studies have shown only indirect links between the PDO with sea level pressure and salinity (Crueger et al., 2009), hydrological balance (Zinke et al., 2008) and river runoff (Grove et al., 2013) in the western Indian Ocean. In addition, our record includes a Sr/Ca record, which is currently considered the most reliable proxy for SST in corals. The only long record from this region of the Indian Ocean is a stable isotope record from Reunion Island that mainly records salinity variations. Therefore, our**

new proxy record from Rodrigues for the period between 1945 and 2006 is a valuable addition to the sparse Indian Ocean coral proxy network. Furthermore, the Cabri records shows statistically significant correlations with the Subtropical Indian Ocean Dipole (SIOD), a fact that we had overlooked in the previous version of the manuscript. We now clearly articulate the climatic link with the SIOD and PDO in the Introduction, Results and Discussion, supported by a number of new references. Our results also demonstrate that records from Rodrigues are well suited when studying climate teleconnections with the SIOD and PDO (as stated in abstract and conclusions). Even the long record might be proven invaluable as a subtropical Indian Ocean record in the near future. We demonstrate that the Totor record does follow grid-SST in the 19[th] and early 20[th] century for several decades. Only further replication with Sr/Ca records from the same site or nearby sites can provide further validation of the long Totor Sr/Ca record. We have also included a new Figure 9 that illustrates the agreement with another coral Sr/Ca-SST proxy record from St. Marie Island off east Madagascar (Grove et al., 2103a) and a Supplementary Figure 2 that illustrates the agreement/disagreement of Totor SST with the longest coral SST reconstruction from the tropical western Indian Ocean (MAHE, Seychelles: Pfeiffer and Dullo, 2006).

Crueger, T., Zinke, J. and Pfeiffer, M. 2009. Dominant Pacific SLP and SST variability recorded in Indian Ocean corals. International Journal of Earth Sciences 98, Special Volume. doi:10.007/s00531-008-0324-1.

Grove, C. A., Zinke, J., Peeters, F., Park, W., Scheufen, T., Kasper, S., Randriamanantsoa, B., McCulloch, M. T. and Brummer, GJA 2012. Madagascar

**corals reveal Pacific multidecadal modulation of rainfall since 1708.** *Climate of the*

*Past* **9, 641-656.**

**Zinke, J., Timm, O., Pfeiffer, M., Dullo, W.-Chr., Kroon, D. and Thomassin, B. A.**

**2008. Mayotte coral reveales hydrological changes in the western Indian between**

**1865 to 1994. Geophysical Research Letters 35, L23707,**

**doi:10.1029/2008GL035634.**

Line 864 (Table 1): clarify that depth is of the colony; also provide time periods for the calculations of average growth characteristics.

**Done. Time periods now indicated.**

Line 882: 'Rodrigues Island'

**Done.**

Lines 887-889: provide the sampling resolution for these graphs.

**Now indicated.**

Lines 892-894: Indicate in Figure caption that y-scale for Sr/Ca is inverted.

**Done.**

Lines 936-939: There are 3 panels to Fig A1 – explain each in caption; also indicate temporal resolution of time series.

**Now indicated.**

Supplementary Tables 1-26: This is a lot of information that I feel needs to be better synthesised for the reader. Also, in carrying out so many correlations on the same set of time series, has any allowance been made for loss of degrees of freedom? i.e. the number of correlations undertaken increases the probability of obtaining a significant correlation by chance.

**In our opinion, it is extremely important to provide the correlation tables with various SST products that all have their own strength' and weaknesses. SST products for the Southern Hemisphere are more strongly affected by measurement biases than Northern hemisphere data, as clearly stated in Jones (2016). We aim to be transparent in showing the individual correlations with**

**various SST products. The reader can infer which SST products shows statistically significant correlations with our individual coral records. All correlations with the coral composite were omitted since we no longer attempt to composite the two core records.**

**For our interpretations, we do not 'pick' a few significant correlations from a large set of linear regressions (which would indeed increase the risk of correlations 'by chance'). It is clear from Supplementary Tables 1-21 that our interpretations are solely based on robust correlation results, i.e. for core Cabri, correlations with grid SST are always strong and significant, regardless of the SST product or season, while for core Totor they are not.**

**Jones, P. The Reliability of Global and Hemispheric Surface Temperature**

**Records. Advances in Atmospheric Sciences, 33, 269-282, 2016.**

**Anonymous Referee #2**

General comments:

This paper presents two new Sr/Ca-SST reconstructions from Rodrigues Island in the south-central Indian Ocean, which contribute to our understanding of SST variability and trends in this region and their relationship to modes of climate variability (PDO, ENSO). The authors present a very rigorous assessment of the potential impact of diagenesis, corallite orientation, and Sr/Ca-SST calibration on these Sr/Ca-SST recon- structions. The results of this analysis emphasize the importance of corallite orientation and screening for diagenesis in such reconstructions, as suggested in previous work by DeLong et al. (2012), Hendy et al. 2007 (and others). However, a number of warm and cold anomalies may not be completely explained by either corallite angle or diagenesis, and the authors conclude that the SST reconstruction is only reliable back to 1945. This leaves little gained beyond the instrumental record, though additional support for the importance of these issues is still an important contribution on its own.

Nonetheless, I have a number of other major concerns that need to be addressed by the authors before publication:

1) Calculation of the composite: the composite was calculated by taking the arithmetic mean of the coral records from each site, yet the authors do not demonstrate strong agreement between the two coral records before compositing. The authors need to show statistics supporting the agreement between the records. E.g., what is the correlation between the two records? Based on figure 3, there appears to be disagreement between the two records such that when averaged, the variability of the composite is reduced over the interval that included the two records (relative to that of the earlier period when only 1 record is available). The two records also have opposing trends over the 1951-2005 interval (as discussed on lines 359-363)! The moderate trend of the composite (0.44 degrees) is simply a result of averaging the strong positive (1.38 degrees) and moderate negative trend (-0.49 degrees) and thus isn't physically interpretable. The climate signal also appears to be weakened in the composite (e.g., Figure A5).

**We have omitted all correlations using the coral composite since we will no longer attempt to compute a composite record.**

2) Selection of Sr/Ca-SST calibration: the authors compute a local calibration with both in situ and gridded SST data, but then use the relationship from Corrège 2006. The justification for this is not clear from the paper. Since local SST data is available, the authors should use this calibration unless they have a valid reason not to use the in situ data.

**We compute calibrations with local and regional grid-SST data over a short time interval 2002 to 2006 and with satellite SST/grid-SST back to 1981 (Table A2). The local calibration with *in-situ* SST is based on only four years and it is currently not known if the slope of the short calibration period would be stable over longer periods. The application of the regression slope for the entire record is therefore not robust. We still find a relatively large spread of Sr/Ca-SST relationships depending on the coral core and the SST record. However, the range of this spread is consistent with the results of Correge, 2006, who used a much larger set of coral Sr/Ca records. Therefore, our regression equations are a confirmation of Correge's work. In addition, we consider the mean Sr/Ca-SST slope of Correge (who used more than 30 coral Sr/Ca records from various ocean basins and different coral genera) to be much more reliable than our short in situ calibration. Corrège (2006) provided regression slopes from a greater density of calibrated records across the global tropics and his 'global' slope of ~ -0.06mmol/°C agrees with the range of slopes that**

**we obtained (Table A2). Most coral records used in Corrège (2006) were calibrated with satellite or grid-SST. Since we are interested in the reconstruction of large-scale southern Indian Ocean SST and their teleconnections with global climate modes, we use of grid-SST. We account for the full spread in regression slopes reported in the literature (-0.4 to -0.084 mmol/°C) and our uncertainty bounds are conservative estimates. In addition, it has been shown by Nurhati et al. (2011) that reconstructions of absolute SST have large errors (up to 7°C) while those of relative SST (anomalies) are lower (<1°C). Therefore, in our study we use SST anomalies calculated with the mean slope from Corrège (2006) for the assessment of interannual climate relationships.**

**Corrège, T., Sea surface temperature and salinity reconstruction from coral geochemical tracers. Palaeogeo. Palaeoclim. Palaeoeco., 232, 408-428, 2006.**

**Nurhati, I. S., K. M. Cobb and E. D. Lorenzo (2011). Decadal-Scale SST and Salinity Variations in the Central Tropical Pacific: Signatures of Natural and Anthropogenic Climate Change. Journal of Climate 24: 3294-3308.**

3) Sr/Ca-SST calibration methodology: the authors should use a reduced major axis regression instead of simple linear regression to calibrate their Sr/Ca records with SST. RMA takes into account errors in both SST and Sr/Ca, which is critical given that the SST observations themselves are also imperfect (see Solow and Huppert 2004; York et al. 2004; Thirumalai et al. 2011). It is also unclear why the authors use only the max and min to calibrate, rather than the full record of monthly anomalies over the calibration period. Justification for this choice is needed.

**Since the work of Solow and Huppert has been cited incorrectly in a couple of coral papers, we report below their original text which discusses the problems of RMA regression (which, according to Solow and Huppert, cannot be solved!).**

**We decided to use ordinary least squares (OLS) regression for the calibration of our coral records, as this is the method best suited for asymmetric relationships. The coral Sr/Ca-SST**

**relationship is clearly asymmetric (SST influences coral Sr/Ca, coral Sr/Ca has no influence on SST). A potential error in the instrumental data does not justify the use of RMA. See Smith (2009): Use and misuse of the reduced major axis for line-fitting (DOI: 10.1002/ajpa.21090) for a discussion.**

**Solow and Huppert (2004) (incorrectly cited by the reviewer as suggesting RMA regression for coral calibrations) also advocate the use of OLS for the calibration of coral proxies. They do not recommend RMA regression:**

**The biggest problems with the application of RMA for coral-Sr/Ca calibrations are the unknown errors. RMA assumes that the error variance in the SST observations equals the error variance of the Sr/Ca determinations. There is no reason to believe that this assumption is warranted. The RMA method can be extended to allow for differences in the error variances. To do so, it is necessary to have an estimate of both the SST and Sr/Ca error variance. However, it is practically impossible to determine the error variance of coral Sr/Ca determinations, as these include not only the analytical error but also other factors such as vital effects or skeletal heterogeneities.**

**Nevertheless, we do not reconstruct absolute SST for the entire time series. Instead we reconstruct relative SST changes or SST anomalies, which have a much lower error than absolute SST estimates (see Nurhati et al., 2011). The calibration exercise is reported in order to give the reader an idea how well absolute SST is recorded for the 4 years of *in-situ* SST measurements. We report the various calibrations since this is now standard procedure.**

4) Potential warm and cool biases in the coral records: It is unclear which of the warm and cool biases (highlighted in figure 6 and discussed on lines 414-424) were included in the composite, and which were removed. From the composite of figure 6, it looks like many of them were averaged in. Clearer justification of their inclusion is needed. Given the magnitude of these anomalies and the fact that their source is unclear (in the cases where no clear diagenesis was identified), the authors should investigate whether removing these events from the record changes their results.

**We have now focused our climate interpretation on the Cabri record and no longer attempt to compute a composite record.**

Specific comments:

Lines 289-292: it is difficult to see this comparison of seasonality from figure 3. I suggest showing the period of overlap separately to demonstrate the agreement between the records

**The period of overlap was illustrated in former Figure 7. Here, in our new Figure 2 we aim to show the entire records.**

Lines 301-303: Figure 3 does not effectively portray the trends discussed here

**We have removed the trends from the text and discuss SST later in the validation section.**

Lines 325-327: Discuss these calibration methods earlier (when discussing the calibration approach in the methods section)

**We have restructured the Results section. We now present the calibration results after the diagenesis section.**

Line 329: This validation period includes part of the calibration period. Stop in 2002 to have independent calibration/validation periods.

**The validation was performed with long term SST and air temperature products only, while the *in-situ* calibration was made with local SST and other products between 2002 and 2006 only. Since we do not reconstruct absolute SST, a strict calibration/validation exercise has not been undertaken. We performed the calibration in order to obtain the spread in regression slopes for the Sr/Ca-SST relationship for the short period of *in-situ* observations. However, we decided to use the mean slope of the Sr/Ca-SST relationship from Corrège (2006) who provided regression slopes from a greater density of calibrated records across the entire tropics.**

Lines 374-394: This comparison with SST over the past 150 years is not a very useful exercise given the paucity of data at this site (as shown in figure A2). The authors seem to be using the agreement with the instrumental data over the full record to support their reconstruction, but this reasoning is circular (we need a coral reconstruction because there aren't enough observations, but then we use the observations to validate our record). Stick to the well observed period for the calibration/validation exercise. It is very possible that some of these discrepancies between the Sr/Ca-SST and SST are due to biases in the SST record.

**We agree with reviewer 2 that that discrepancies between proxy and SST data simply arise from a lack of SST observations. At present, it is not clear which gridded SST data are most suited for the Indian Ocean and tropical oceans in general. The use of multiple SST products is now a standard procedure in almost all meteorological studies. We have adopted this approach in our manuscript. In our opinion it is therefore extremely important to assess/illustrate the agreement between various SST products for our region with our proxy data. Currently, this is the only independent method to assess which SST products might perform better in specific ocean basins. We have also included a new Figure 9 that illustrates the agreement with another coral Sr/Ca-SST proxy record from St. Marie Island off east Madagascar (Grove et al., 2103a).**

Line 412: see also Sayani et al. 2011

Lines 412-413: What about dissolution, any indication that dissolution could explain these discrepancies (e.g., see Sayani et al. 2011) who show that dissolution is associated with low Sr/Ca anomalies/warm biases)?

**Thin sections and SEM studies are often used to detected dissolution in reef corals (Hendy et al., 2007; McGregor and Abram, 2008; Sayani et al., 2011). The application of both techniques in this study showed that the two coral cores are devoid of dissolution. Hendy et al (2007) showed that dissolution during marine diagenesis leads to an increase in Sr/Ca and therefore an apparently cold temperature anomaly. Dissolution during marine diagenesis therefore would not be able to cause the observed positive temperature anomaly. Decreased Sr/Ca values in diagenetically modified corals have been attributed to aragonite**

**dissolution and concomitant calcite cementation in a meteoric environment (Sayani et al., 2011). With a combination of SEM, thin section microscopy and XRD we demonstrated the lack of dissolution and calcite cementation in the corals and therefore can rule out any influence of dissolution on the proxy record.**

**Hendy, E. J., Gagan, M. K., Lough, J. M., McCulloch, M., and deMenocal P. B.: Impact of skeletal dissolution and secondary aragonite on trace element and isotopic climate proxies in Porites corals, Paleoceanography, 22, PA4101, doi:10.1029/2007PA001462, 2007.**

**McGregor, H. V. and Abram, N. J.: Images of diagenetic textures in Porites corals from Papua New Guinea and Indonesia, Geochemistry, Geophysics, Geosystems 9(10), doi:10.1029/2008GC002093, 2008.**

**Sayani, H. R., Cobb, K. M., Cohen, A. L., Crawford Elliott, W., Nurhati, I. S., Dunbar, R. B., Rose, K. A., Zaunbrecher, L. K.: Effects of diagenesis on paleoclimate reconstructions from modern and young fossil corals, Geochimica et Cosmochimica Acta, 75, 6361–6373, 2011.**

Lines 434-436: Recommend performing a running correlation analysis to test this.

**No longer applicable. The section in question referred to non-stationary relationships between the coral composite and large-scale SST. We no longer use the composite as requested by the reviewers. Instead, we now focus our climate interpretation on the Cabri record that extends from 1945 to 2006.**

Lines 522-526: other potential drivers of these discrepancies? E.g., see Alpert et al. 2015

**We mentioned unknown vital effects as a potential driver of the discrepancies.**

Line 602-607: corals may have acclimatized or adapted to the high temperature variability. A number of studies have shown that corals in sites that have high temperature variability may be less susceptible to bleaching (e.g., Thompson and van Woesik 2009, Donner 2011). This variability is now usually taken into account when calculating the thermal stress thresholds to predict bleaching (e.g., Kleypas et al. 2015), as this approach has been shown to better predict observed bleaching patterns (e.g., Logan et al. 2012); this should be used instead of the conventional degree heating weeks threshold on lines 605-607.

**We have deleted this section.**

Figure 8 caption: this caption needs to be reworded. It is hard to follow what is in each panel.

**Done.**

Figure 9: why divide the analysis into these periods? This needs to be justified some- where. If the goal was to compare among different phases of the PDO & ENSO (which from the text appears to be the goal), then the authors should select periods that line up with the phases of these modes.

**We divided the analysis into different periods in order to test for differences in spatial correlation patterns and their stability over different time series length. We used the full overlap period with grid-SST of the Cabri dataset and two multi-decadal sub-periods. For instance, the 1961-1990 period is chosen to use SST data from a period that includes pre- and post-satellite era observations, yet does not include the most recent years between 1991 and 2006.**

Figure 2: change the color scheme so that the lines in 2a are differentiable

**We changed the color scheme.**

Figure 7: the markings denoting corallite angle are not clear, where do the transitions occur, at the end of the lines? May be clearer if brackets are used to denote the sections with different corallite angles

**We now use brackets instead of colors.**

Figure A1: change the color scheme so that the black lines are differentiable

**We changed the color scheme.**

Figure A4: what is the difference between the figures on the left and right? This is not indicated in the caption, and it is not clear from the figure. Clarify in caption.

**Done. This Figure is now Figure 8 in the main text.**

Technical corrections:

Lines 384-387: reword

**Done.**

Line 432: long-term

**Done.**

Line 548: closest agreement to Table A1 caption, line 977: change "in brackets" to "in parentheses"

**Done.**

[revised manuscript text omitted]

Generally diagenesis could be excluded as a major cause of discrepancies between coral SST and grid-SST. For core Totor, only for the period between 1882 and 1887 is diagenesis a potential cause of a cool bias on our coral SST reconstruction (Figure 3d). Core Cabri showed only localized positive Sr/Ca anomalies (cool SST bias) caused by aragonitic sediment trapped within growth framework pores (Fig. 3f). These specific samples have been removed before interpolation. Having excluded diagenesis for almost all of the record, we assessed sampling biases due to changes in the orientation of growth axes and positioning of corallites to the slab surface (Tab. 2 & 3). De Long et al. (2012) showed clear evidence for warm or cool biases in coral Sr/Ca-SST reconstructions caused by suboptimal orientation of corallites in corals from New Caledonia. We have adopted a similar approach to test for sampling biases in our two cores (summarized in Table 2 & 3). We found that core Totor contained areas where a sampling bias could explain anomalous Sr/Ca-derived SST (1781-1797, 1825-1835, 1854-1860, 1916-1921, 1936-1941 and 1948-1951, 1984-2001). We provide a detailed explanation of the potential biases in core Totor in the Supplementary Information that is of particular importance for coral paleoclimatologists.

De Long et al. (2012) showed that warm biases were often caused by corallites orientated at an angle or oblong to the slab surface and where growth orientation had
* * *
Jens Zinke 16/8/2016 4:34 PM

Jens Zinke 16/8/2016 4:35 PM
**Comment [21]:** Discussion shortened and focussed on main findings

Davies, G.R. 12/8/2016 11:19 AM

Davies, G.R. 12/8/2016 11:20 AM

Davies, G.R. 12/8/2016 11:20 AM

Jens Zinke 16/8/2016 4:36 PM
**Comment [22]:** Here, an entire paragraph has been moved to Supplementary Information.

Davies, G.R. 12/8/2016 11:21 AM

[revised manuscript text omitted]

* * *
Davies, G.R. 12/8/2016 11:33 AM

Davies, G.R. 11/8/2016 10:16 PM

Jens Zinke 16/8/2016 4:56 PM

Davies, G.R. 12/8/2016 11:34 AM

Jens Zinke 16/8/2016 4:56 PM

Jens Zinke 16/8/2016 4:56 PM

Davies, G.R. 12/8/2016 11:34 AM

Jens Zinke 16/8/2016 4:56 PM

Jens Zinke 16/8/2016 4:56 PM

Davies, G.R. 12/8/2016 11:34 AM

Jens Zinke 16/8/2016 4:56 PM

Jens Zinke 16/8/2016 4:56 PM

Jens Zinke 16/8/2016 4:56 PM

Jens Zinke 16/8/2016 4:56 PM

Davies, G.R. 12/8/2016 11:35 AM

Jens Zinke 16/8/2016 4:56 PM

Jens Zinke 16/8/2016 4:56 PM

Davies, G.R. 12/8/2016 11:35 AM

Jens Zinke 16/8/2016 4:56 PM

Jens Zinke 16/8/2016 4:56 PM

Davies, G.R. 12/8/2016 11:35 AM

Jens Zinke 16/8/2016 4:56 PM
Comment [25]: new and modified paragraph

Jens Zinke 16/8/2016 4:56 PM

Davies, G.R. 11/8/2016 10:16 PM

[revised manuscript text omitted]

Formatted ... [10]
Davies, G.R. 11/8/2016 10:16 PM
Formatted ... [11]
Davies, G.R. 11/8/2016 10:16 PM
Formatted ... [12]
Davies, G.R. 11/8/2016 10:16 PM
Formatted ... [13]
Davies, G.R. 11/8/2016 10:16 PM
Formatted ... [14]
Davies, G.R. 11/8/2016 10:16 PM
Formatted ... [15]
Davies, G.R. 11/8/2016 10:16 PM
Formatted ... [16]
Davies, G.R. 11/8/2016 10:16 PM
Formatted ... [17]
Davies, G.R. 11/8/2016 10:16 PM
Formatted ... [18]
Davies, G.R. 11/8/2016 10:16 PM
Formatted ... [19]
Davies, G.R. 11/8/2016 10:16 PM
Formatted ... [20]
Davies, G.R. 11/8/2016 10:16 PM
Formatted ... [21]
Davies, G.R. 11/8/2016 10:16 PM
Formatted ... [22]
Davies, G.R. 11/8/2016 10:16 PM
Formatted ... [23]
Davies, G.R. 11/8/2016 10:16 PM
Formatted ... [24]
Davies, G.R. 11/8/2016 10:16 PM
Formatted ... [25]
Davies, G.R. 11/8/2016 10:16 PM
Formatted ... [26]
Davies, G.R. 11/8/2016 10:16 PM
Formatted ... [27]
Davies, G.R. 11/8/2016 10:16 PM
Formatted ... [28]
Davies, G.R. 11/8/2016 10:16 PM
Formatted ... [29]
Davies, G.R. 11/8/2016 10:16 PM
Formatted ... [30]
Davies, G.R. 11/8/2016 10:16 PM
Formatted ... [31]
Davies, G.R. 11/8/2016 10:16 PM
Formatted ... [32]
Davies, G.R. 11/8/2016 10:16 PM
Formatted ... [33]
Davies, G.R. 11/8/2016 10:16 PM
Formatted ... [34]
Davies, G.R. 11/8/2016 10:16 PM
Formatted ... [35]
Davies, G.R. 11/8/2016 10:16 PM
Formatted ... [36]

[revised manuscript text omitted]

---

## Author Response (AR2)

Institut für Geowissenschaften,
Sektion Paläontologie,
Malteserstrasse 74-100,
12249 Berlin, Germany

Freie Universität Berlin, Institut für Geowissenschaften,
Section Paleontology
Malteserstrasse 74-100, 12249 Berlin

To
Editor Biogeosciences
Natascha Töpfer
Copernicus Publications

Dr. Jens Zinke
Sektion Paläntologie
12249 Berlin

**Telefon**  +49 30 838-61034
**Fax**  +49 30 838-70745
**E-Mail**  jzens.inke@fu-berlin.de
**Internet**  www.fu-berlin.de

04.10.2016

**Revision bg-2016-69**

Dear Editor,

Please find enclosed our final corrections for manuscript bg-2016-69 entitled "A sea surface temperature reconstruction for the southern Indian Ocean trade wind belt from corals in Rodrigues Island (19°S, 63°E)", for publication in Biogeosciences.

We thank both reviewers and you as Editor for the constructive comments that helped us to improve the mansucript and the acceptance of our paper. Below, you will find our detailed response to the specific comments by reviewer 1.

We sincerely hope that our revised mansucript is now suitable for publication in Biogeosciences.

Kind regards,
Jens Zinke

Answer to Specific comments:

Line 62: 'Synchronously'
Done.

Line 64: Figure 1b referenced but Figure 1a should come first.
Corrected.

Line 92: 'coral paleo-thermometers'
Done.

Line 93: 'alteration'
Done.

Line 99: 'Subtropical'
Done.

Line 154: 'occurring mostly'
Done.

Line 176: 'Fig. 1a'
Done.

Line 176-177: 'colony'
Done.

Line 179: 'August 2005 from a colony on the………….'
Done.

Line 181: 'March 2007 from a colony growing……..'
Done.

Line 191: 'Figs. A4 and A5' – should be relabelled/re-ordered in SM as Fig. A1 and Fig. A2 – as first occurrence in text.
Corrected.

• OK just realised my confusion between material presented in an Appendix and Supplementary material. Should these not be combined into one Supplementary Material file to prevent other readers being similarly confused.
We like to keep the Appendix figures separated from the Supplements because we feel that the Appendix figures provide important detail. The journal allows for Appendix figures to be separated from Supplements.

Line 191: 'coral density was calculated'
Done.

Line 196: 'samples for geochemical analysis'
Done.

Lines 202-204: 'Corals that showed an…………..'; given there are only 2 corals, do the authors mean sections of corals?
Corrected.

Line 271-208: 'along the geochemical sampling tracks'
Done.

Line 221: 'was dissolved'
Done.

Line 222: 'was diluted'
Done.

Line 250: 'extracted various SST and….'
Done.

Line 251: 'for comparison with'
Done.

Line 257: HadISST is not listed in Table A1
Corrected.

Line 264: HadSST3 also not listed in Table A1
Corrected.

Line 269: HadMAT1 and HadnMAT2 not listed in Table A1
Corrected.

Line 273: Section 3 Materials and Methods should also include a description of the statistical analyses undertaken; the section should also include the description and data sources of the climate indices used (e.g. SIOD, PDO, ENSO) and other proxy coral records used.
Now included.

Line 282: 'between the two colonies'
Done.

Line 283: As Figure 2a presents Sr/Ca monthly anomalies, it is not possible to see the 'distinct seasonality'referred to in the text.
Corrected.

Line 288: anomalies 'higher' than what?
Relative to 1961-1990, now indicated.

Line 291: 'seasonality' not evident in Fig. 2a
Corrected.

Lines 309-310: Figure 4 does not show a radiograph
Corrected.

Line 312: 'and therefore the reported…'
Done.

Lines 334-335: Explain why just these 3 data sets were used for calibration.
Now reads "since SST seasonality does not differ significantly between SST products for the 2002 to 2006 period"

Lines 351-406: Section 5.4 – I still find this section extremely hard to follow and identify the key findings of the authors.
Without clearer definition what is hard to follow we cant address this point.

Line 406: I think that part of section 6.1 should appear in the Results rather than the Discussion. Need to clearly separate the findings of the study from discussing their implications etc.
We do not agree.

Line 410: Reference to Figures 7 and 8 in text before Figure 6.
Changed order of Figures.

Line 415: 'shows significant positive'; elsewhere referring to correlations indicated whether significant.
Done.

Lines 436-437: Figure 9 does not appear to show the Reunion Island record referred to.
Corrected.

Line 478: First reference to Figure 6 in text!
Order of Figures changed.

Line 481: 'very low overall density' – compared to what?
"compared to pre-1984 record" added

Line 506: 'new cores need to be obtained………'
Done.

Line 844: 'density'
Done.

Line 989: 'inverted' rather than' converted'

Done.

Lines 1000-1009: Please clarify labelling and caption to make it clear to reader which core sample is represented in each panel (e.g. Totor or Cabri).
Corrected.

Line 1030: How was this image of the coral core obtained? Is it under UV light?
No, just image software with blue colour.

Pages 54 and 55: Are these really positive X-ray prints, they look more like X-ray negatives?
Corrected.

[revised manuscript text omitted]

Oldenborgh and Burgers, 2005).

[Figure]

[Figure]

[Figure]

[Figure]

Jens Zinke 4/10/2016 4:34 PM

Jens Zinke 4/10/2016 4:35 PM

Jens Zinke 4/10/2016 4:35 PM

[Figure]

**a**

corr Jul-Jun averaged HadISST 63E, 19S
with Jul-Jun averaged HadMAT1 Tair (detrend), 1945-2001, p<0.05

[Figure]

**b**

corr Jul-Jun averaged Cabri SST
with Jul-Jun averaged HadMAT1 Tair (detrend), 1945-2001, p<0.05

Figure A4 – a) Spatial correlation of HadISST for the grid box of Rodrigues Island with global HadMAT1 marine air temperature (Tair) between 1945 and 2001 for July to June annual averages (Rayner et al., 2003). Note the location of Rodrigues Island marked by yellow star. b) same as a), but for Cabri SST with global HadMAT1. Only correlations with p<0.05 coloured. Computed at knmi climate explorer (van Oldenborgh and Burgers, 2005).

Jens Zinke 4/10/2016 4:34 PM

Jens Zinke 5/10/2016 12:19 PM

Jens Zinke 5/10/2016 12:19 PM

Jens Zinke 5/10/2016 12:21 PM

Jens Zinke 5/10/2016 12:20 PM

Jens Zinke 5/10/2016 12:20 PM

Jens Zinke 5/10/2016 12:20 PM

Jens Zinke 5/10/2016 12:20 PM

Jens Zinke 5/10/2016 12:20 PM

Jens Zinke 4/10/2016 4:33 PM

|                | **SST *in situ*** **2002-2006** | **AVHRR SST** **2002-2006** | **ERSST** **2002-2006** | **Air Temp.** **2002-2006** |
|----------------|---------------------------------|------------------------------|--------------------------|------------------------------|
| Mean annual    | 25.49 (0.24)                    | 25.4 (0.11)                  | 25.57 (0.3)              | 27.49 (0.31)                 |
| Maximum        | 28.6 (0.5)                      | 28.65 (0.44)                 | 28.29 (0.4)              | 31.2 (0.62)                  |
| Minimum        | 22.4 (0.27)                     | 22.75 (0.21)                 | 23.15 (0.13)             | 24.2 (0.44)                  |
| Seasonal Range | 6.22 (0.68)                     | 5.9 (0.58)                   | 5.14 (0.39)              | 7.0 (0.79)                   |
| STDV           | 2.14                            | 1.78                         | 1.69                     | 2.07                         |

Table A1 – Statistics of various sea surface temperature (SST) products and air temperature for Rodrigues with $1\sigma$ standard deviations in brackets for the period 2002 to

2006 (period with *in situ* SST data). STDV = $1\sigma$ standard deviation over all years. All units in °C.

| (a) Max-Min | Regression equation | r$^2$ | p |
|---|---|---|---|
| Totor | Sr/Ca = -0.0439(±0.004)*SST + 10.032(±0.10) | 0.97 | <0.001 |
| Cabri | Sr/Ca = -0.0384(±0.005)*SST + 9.861(±0.12) | 0.89 | <0.001 |
| (b) Max-Min | | | |
| Totor | Sr/Ca = -0.0638(±0.004)*SST + 10.566(±0.09) | 0.95 | <0.001 |
| Cabri | Sr/Ca = -0.0507(±0.004)*SST + 10.179(±0.10) | 0.90 | <0.001 |
| (c) Max-Min | | | |
| Totor | Sr/Ca = -0.0531(±0.004)*SST + 10.271(±0.11) | 0.96 | <0.001 |
| Cabri | Sr/Ca = -0.0441(±0.005)*SST + 10.012(±0.13) | 0.88 | <0.001 |
| (d) Monthly | | | |
| Totor | Sr/Ca = -0.0522(±0.003)*SST + 10.272(±0.08) | 0.79 | <0.001 |
| Cabri | Sr/Ca = -0.0419(±0.003)*SST + 9.95(±0.07) | 0.87 | <0.001 |

Table A2 - Linear regression of coral Sr/Ca with a) *in situ* SST 2002-2005/6, b)

ERSSTv.3 (Smith et al., 2008) 1997-2005/6, c) AVHRR SST NOAA Coral Reef watch data 2000-2005/6 and d) monthly Sr/Ca with AVHRR SST (Reynolds et al., 2007) for the period 1982 to 2005.